# Nonparametric Identifiability of Causal Representations from Unknown Interventions

**Julius von Kügelgen**[1,2], **Michel Besserve**[1], **Liang Wendong**[1], **Luigi Gresele**[1], **Armin Kekić**[1],
**Elias Bareinboim**[3], and **David M. Blei**[3], and **Bernhard Schölkopf**[1]

[1]Max Planck Institute for Intelligent Systems, Tübingen, Germany
[2]Department of Engineering, University of Cambridge, United Kingdom
[3]Columbia University, USA

## Abstract

We study causal representation learning, the task of inferring latent causal variables and their causal relations from high-dimensional functions ("mixtures") of the variables. Prior work relies on weak supervision, in the form of counterfactual pre- and post-intervention views or temporal structure; places restrictive assumptions, such as linearity, on the mixing function or latent causal model; or requires partial knowledge of the generative process, such as the causal graph or intervention targets. We instead consider the general setting in which both the causal model and the mixing function are *nonparametric*. The learning signal takes the form of multiple datasets, or environments, arising from *unknown interventions* in the underlying causal model. Our goal is to identify both the ground truth latents and their causal graph up to a set of ambiguities which we show to be irresolvable from interventional data. We study the fundamental setting of two causal variables and prove that the observational distribution and one perfect intervention per node suffice for identifiability, subject to a genericity condition. This condition rules out spurious solutions that involve fine-tuning of the intervened and observational distributions, mirroring similar conditions for nonlinear cause-effect inference. For an arbitrary number of variables, we show that at least one pair of distinct perfect interventional domains per node guarantees identifiability. Further, we demonstrate that the strengths of causal influences among the latent variables are preserved by all equivalent solutions, rendering the inferred representation appropriate for drawing causal conclusions from new data. Our study provides the first identifiability results for the general nonparametric setting with unknown interventions, and elucidates what is possible and impossible for causal representation learning without more direct supervision.

## 1 Introduction

Causal representation learning (CRL) seeks to describe high-dimensional, low-level observations through a small number of interpretable, causally-related latent variables [108, 110]. In doing so, its goal is to combine the strengths of classical causal inference with those of modern machine learning. A causal model represents an entire family of distributions arising from *interventions* on a system of variables [10, 95, 100]. This provides a principled way for reasoning about distribution shifts, which facilitates out-of-distribution generalization and planning [9, 64, 96, 102]. However, causal models require that most (or at least some) relevant causal variables are directly observed. While reasonable in domains such as economics [7], social [91] or biomedical science [48, 57], this assumption has challenged the application of causal methodology to complex and high-dimensional data [85]. Machine learning, on the other hand, has proven successful at learning useful "representations"—latent vectors generating the observables via some nonlinear map—of high-dimensional data such as images, video, or text [16, 20, 77, 106]. However, most methods rely on independent and identically distributed (i.i.d.) data and only extract *associational* information. As a consequence, they often fail under distribution shifts and do not generalize beyond the training distribution [107], as exhibited by their reliance on *spurious correlations* [13, 87] or their vulnerability to *adversarial examples* [40, 119].

37th Conference on Neural Information Processing Systems (NeurIPS 2023).

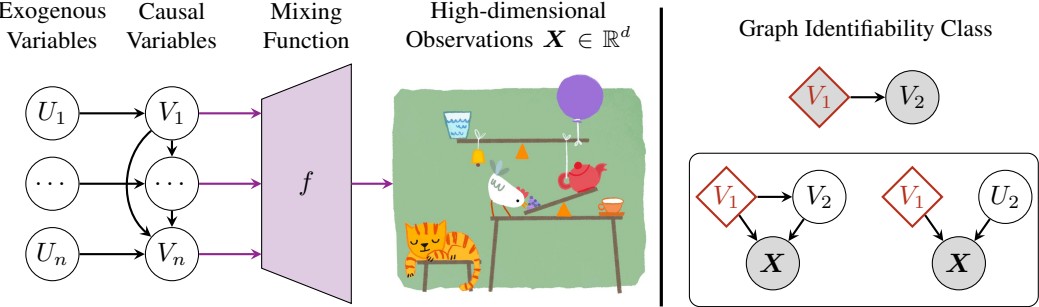

Figure 1: *(Left)* **Data-Generating Process for Causal Representation Learning (CRL).** Observations $\boldsymbol{X}$ are generated by applying a nonlinear mixing function $f$ to a set of causal latent variables $\boldsymbol{V} = \{V_1, ..., V_n\}$, which are related through a structural causal model (SCM) with independent exogenous variables $\boldsymbol{U} = \{U_1, ..., U_n\}$; illustration by Ana Martín Larrañaga. *(Right)* **Comparison to Structure Learning.** The causal direction between two unconfounded *observed* variables (top) is uniquely identified from a single intervention [37]; for CRL (bottom), this is not the case, as the nonlinear mixing introduces additional ambiguity due to spurious representations. Shaded nodes are observed, white ones unobserved, and interventions highlighted as red diamonds.

**Identifiability.** It has been argued that to address these shortcomings, we ought to learn representations endowed with causal model semantics. However, a major challenge to this goal is that *different representations can explain the same data equally well*. Strictly simpler representation learning tasks, such as disentanglement [14] or independent component analysis (ICA) [53], are already *non-identifiable* in general [54, 83]. Identifiability studies are thus required to characterize additional assumptions under which the desired latent variables can be provably recovered [4, 17, 21, 31, 42, 43, 45, 46, 51, 52, 55, 56, 66, 69, 70, 75, 76, 84, 90, 101, 112, 114, 127, 141]. In CRL, we need not only identify the latents, but also the causal graph encoding their relations. Even in the fully observed case, this task of causal discovery, or structure learning, is very challenging: the graph can only be recovered up to Markov equivalence [115] based only on (observational) i.i.d. data [116], meaning that the direction of some edges cannot be determined. For CRL, the task gets strictly harder. For instance, if the causal latents $V_i$ in Fig. 1 *(Left)* form a valid representation, then replacing them by the *independent* exogeneous $U_i$ might be considered an equally valid alternative.

**What sets *causal* representations apart?** A crucial feature of causal variables is that they are the ones on which *interventions* are defined and whose relations we are interested in [95]. Causal discovery and CRL thus often rely on non-i.i.d. data linked to interventions on the underlying causal variables [59, 72]. Unless all variables are subject to intervention, however, some fundamental differences between the fully observed and representation learning settings in the level of ambiguity in the graph remain [117], as illustrated in Fig. 1 *(Right)*. In a sense, the non-identifiabilities of representation and structure learning combine, and both need to be addressed in conjunction.

**Problem Setting.** We study the general *nonparametric* CRL problem (§ 2.1) in which both the causal mechanisms and the mixing function are completely unconstrained. Our goal (§ 2.2) is to identify the latent causal variables up to element-wise nonlinear rescaling and their graph up to isomorphism (Defn. 2.6). As motivated above, doing so without further supervision requires access to interventional data. To this end, we consider learning from heterogeneous data from multiple related domains, or *environments*, that arise from interventions in a shared underlying causal model (§ 2.3).

**Contributions.** Our main contributions are theoretical in nature (§ 3). First, we establish the minimality of the targeted equivalence class (Prop. 3.1) in the sense that its ambiguities cannot be resolved from interventional data. We then present our main identifiability results. For the case of two latent causal variables, we show that an observational environment and one for each perfect intervention on either variable suffice (Thm. 3.2)—provided that the intervened and unintervened mechanisms are not "fine-tuned" to each other, which we formalize in the form of a *genericity condition* (3.2). For any number of latent causal variables, we prove that access to pairs of environments corresponding to two distinct perfect interventions on each node guarantees identifiability (Thm. 3.4). We then question how to use or interpret causal representations (§ 4), and show that certain quantities, such as the strengths of causal influences among variables, are preserved by all equivalent solutions (Thm. 4.2). We sketch possible learning objectives (§ 5), and empirically investigate training different generative models (§ 6), finding that only those based on the correct causal structure attain the best fit and identify the ground truth. We conclude by discussing limitations and extensions of our work (§ 7).

**Related Work on *Multi-Environment* CRL.** Most closely related are recent identifiability studies which also leverage multiple environments arising from single node interventions [5, 22, 117, 122], thus mirroring in different interventional setups the result of Brehmer et al. [18] based on counterfactual, multi-view data, which is harder to obtain. Squires et al. [117] provide results for linear causal models and linear mixing; Ahuja et al. [5] consider nonlinear causal models and polynomial mixings, subject to additional constraints on the latent support [125]; and Varici et al. [122] employ a score-based approach for nonlinear causal models and linear mixing. The concurrent work of Buchholz et al. [22] extends the results of Squires et al. [117] to general nonlinear mixings and linear Gaussian causal models. Liu et al. [82] leverage recent advances in nonlinear ICA [55, 65] to identify a linear Gaussian causal model with context-dependent weights and nonlinear mixing from sufficiently diverse environments. A more extensive discussion of other related works and a detailed comparison of existing results with the present study are provided in Appx. A and Tab. 1.

## 2 Nonparametric Causal Representation Learning

In this section, we describe the considered problem setting and state our main assumptions. First, we specify the assumed data generating process (§ 2.1) and learning task in the form of a target identifiability class (§ 2.2). We then demonstrate the hardness of our task from i.i.d. data or imperfect interventions and use this to motivate a multi-environment approach with perfect interventions (§ 2.3).

**Notation.** We use upper-case $X$ for random variables and lower-case $x$ for their realizations. Bold uppercase $\boldsymbol{X}$ denotes random vectors and lowercase $\boldsymbol{x}$ their realizations. We assume throughout that all distributions $P_X$ possess densities $p(x)$ with respect to (w.r.t.) the Lebesgue measure. We denote the pushforward of $P_X$ through a measurable function $f$ by $f_*(P_X)$, and write $[n] = \{1, ..., n\}$.

### 2.1 Data Generating Process

The assumed data generating process consists of a latent causal model and a mixing function, see Fig. 1 *(left)*. For the former, we build on the structural causal model (SCM) framework [95, 100].

**Definition 2.1** (Latent SCM). Let $\boldsymbol{V} = \{V_1, ..., V_n\}$ denote a set of causal "endogenous" variables, with each $V_i$ taking values in $\mathbb{R}$, and let $\boldsymbol{U} = \{U_1, ..., U_n\}$ denote a set of mutually independent "exogenous" random variables. The latent SCM consists of a set of structural equations

$$\{V_i := f_i(\boldsymbol{V}_{\mathrm{pa}(i)}, U_i)\}_{i=1}^n. \tag{2.1}$$

where $\boldsymbol{V}_{\mathrm{pa}(i)} \subseteq \boldsymbol{V} \setminus \{V_i\}$ are the direct causes, or causal parents, of $V_i$, and $f_i$ are deterministic functions; and a fully factorized joint distribution $P_{\boldsymbol{U}}$ over the exogenous variables. The associated causal diagram $G$, a directed graph with vertices $\boldsymbol{V}$ and edges $V_j \to V_i$ iff. $V_j \in \boldsymbol{V}_{\mathrm{pa}(i)}$, is assumed acyclic.

By acyclicity, recursive substitution of the assignments in (2.1) yields the reduced form $\boldsymbol{V} = f_{\mathrm{RF}}(\boldsymbol{U})$. The SCM thus induces a unique distribution $P_{\boldsymbol{V}}$ over the endogenous variables, given by the pushforward of $P_{\boldsymbol{U}}$ via (2.1), that is, $P_{\boldsymbol{V}} = f_{\mathrm{RF}*}(P_{\boldsymbol{U}})$. By construction, $P_{\boldsymbol{V}}$ is Markovian w.r.t. the causal graph $G$ [95, 100], meaning that its density obeys the causal Markov factorization:

$$p(v_1, \ldots, v_n) = \prod_{i=1}^n p_i(v_i \mid \boldsymbol{v}_{\mathrm{pa}(i)}). \tag{2.2}$$

We place the following additional assumption on the distribution $P_{\boldsymbol{V}}$ induced by the SCM.

**Assumption 2.2** (Faithfulness). The *only* conditional independence relations satisfied by $P_{\boldsymbol{V}}$ are those implied by $\{V_i \perp\!\!\!\perp \boldsymbol{V}_{\mathrm{nd}(i)} \mid \boldsymbol{V}_{\mathrm{pa}(i)}\}_{i \in [n]}$, where $\boldsymbol{V}_{\mathrm{nd}(i)}$ denotes the non-descendants of $V_i$ in $G$.

Asm. 2.2 ensures a one-to-one correspondence between (conditional) independence in $P_{\boldsymbol{V}}$ and graphical separation in $G$ and is a standard assumption in causal discovery [115]. Faithfulness rules out cancellations of influences along different paths, which occurs with probability zero for random path-coefficients [121]. It can thus also be viewed as a minimality or genericity assumption.

In contrast to classical causal inference, we assume that both the exogenous variables $\boldsymbol{U}$ and the endogenous causal variables $\boldsymbol{V}$ are unobserved. Instead, we will only have access to $d$-dimensional nonlinear mixtures $\boldsymbol{X}$ of $\boldsymbol{V}$. We therefore make the following additional assumption.

**Assumption 2.3** (Known $n$). The number $n$ of latent causal variables is known.

Next, we specify the relationship between the unobserved causal variables $\boldsymbol{V}$ and the observed $\boldsymbol{X}$.

**Definition 2.4** (Mixing function). The observations $\boldsymbol{X}$ are deterministically generated from $\boldsymbol{V}$ by applying a mixing function $f : \mathbb{R}^n \to \mathbb{R}^d$ to $\boldsymbol{V}$, that is, $\boldsymbol{X} := f(\boldsymbol{V})$.

The terminology and setting of a deterministic mixing is rooted in the nonlinear ICA literature [56]. In deep generative models, it is commonly relaxed by considering additive noise in Defn. 2.4 [65, 90]. For representation learning scenarios, we are particularly interested in the case $n \ll d$. To allow for recovery of $\boldsymbol{V}$ from $\boldsymbol{X}$, we assume that $f$ is invertible, which is a standard assumption for identifiability.

**Assumption 2.5** (Diffeomorphic mixing). $f$ is a diffeomorphism[1] onto its image $\operatorname{Im}(f) = \mathcal{X} \subseteq \mathbb{R}^d$.

## 2.2 Learning Target: The CRL Identifiability Class $\sim_{\mathrm{CRL}}$

Our goal is to infer the underlying latent causal variables $\boldsymbol{V} = f^{-1}(\boldsymbol{X})$ and their causal relations. We therefore consider the true unmixing function $f^{-1}$ *and* the causal graph $G$ our joint learning target: $f^{-1}$ informs us how to map observations $\boldsymbol{X}$ to causal variables $\boldsymbol{V}$, and $G$ tells us how to factorise the implied joint $p(\boldsymbol{v})$ into the causal mechanisms $p_i(v_i \mid \boldsymbol{v}_{\mathrm{pa}(i)})$ from (2.2). Given only observations of $\boldsymbol{X}$, this is a challenging task since neither $\boldsymbol{V}$ nor $G$ are directly observed or known a priori.

When is a candidate solution $(h, G')$ that satisfies a given learning objective (such as maximizing the likelihood, possibly subject to additional constraints) guaranteed to match the ground truth $(f^{-1}, G)$? This is the subject of identifiability studies and the main focus of our work. A statistical model $\mathcal{P} = \{p_\theta : \theta \in \Theta\}$ with parameter space $\Theta$ is identifiable if the mapping $\theta \mapsto p_\theta$ is injective [78]. For representation learning tasks, full identifiability is often not attainable, as there are some fundamental ambiguities that cannot be resolved. One therefore typically instead considers identifiability up to an appropriately chosen equivalence class in the model space [2, 65, 127].

For the assumed data generating process (§ 2.1), the order of the causal variables is arbitrary, since $\boldsymbol{V}$ is unobserved. We can therefore assume without loss of generality (w.l.o.g.) that the $V_i$'s are partially ordered w.r.t. $G$, that is, $V_i \to V_j \implies i < j$.[2] Learning $G$ thus reduces to inferring whether the edges $\{V_1, \ldots, V_{i-1}\} \to V_i$ exist for $i = 2, \ldots, n$. The only remaining permutation ambiguity arises from permutations $\pi$ that preserve the partial order: for example, if $G$ is given by $V_1 \to V_3 \leftarrow V_2$, the order of $V_1$ and $V_2$ cannot be uniquely determined without further assumptions. Moreover, the scaling of the causal variables is also arbitrary: any invertible element-wise transformation can be undone as part of $f$. We therefore define the desired identifiability class through the following equivalence relation.[3]

**Definition 2.6** ($\sim_{\mathrm{CRL}}$-identifiability). Let $\mathcal{H}$ be a space of unmixing functions $h : \mathcal{X} \to \mathbb{R}^n$ and let $\mathcal{G}$ be the space of DAGs over $n$ vertices. Let $\sim_{\mathrm{CRL}}$ be the equivalence relation on $\mathcal{H} \times \mathcal{G}$ defined as

$$(h_1, G_1) \sim_{\mathrm{CRL}} (h_2, G_2) \iff (h_2, G_2) = (\boldsymbol{P}_{\pi^{-1}} \circ \phi \circ h_1, \pi(G_1)) \tag{2.3}$$

for some element-wise diffeomorphism $\phi(\boldsymbol{v}) = (\phi_1(v_1), \ldots, \phi_n(v_n))$ of $\mathbb{R}^n$ and a permutation $\pi$ of $[n]$ such that $\pi : G_1 \mapsto G_2$ is a graph isomorphism and $\boldsymbol{P}_\pi$ the corresponding permutation matrix.

*Remark* 2.7. When $G$ has no edges, any permutation is admissible and $\sim_{\mathrm{CRL}}$ reduces to the standard notion of identifiability up to permutation and element-wise reparametrisation of nonlinear ICA [56].

The ground truth $(f^{-1}, G)$ is identified up to $\sim_{\mathrm{CRL}}$ by a given learning objective if any candidate solution $(h, G')$ satisfies $(h, G') \sim_{\mathrm{CRL}} (f^{-1}, G)$. We seek to discover suitable conditions that ensure this.

## 2.3 Multi-Environment Data

Given only a single dataset of i.i.d. observations from $P_{\boldsymbol{X}}$, there is no hope for $\sim_{\mathrm{CRL}}$-identifiability. Even for observed $\boldsymbol{V}$ (i.e., with $n = d$ and known $f = \mathrm{id}$), $G$ can only be identified up to Markov equivalence [115]. With unknown mixing $f$, the degree of observational non-identifiability gets even worse: for example, by using the reduced form of the SCM (§ 2.1) we can express $\boldsymbol{X}$ in terms of the latent exogenous variables $\boldsymbol{U}$ via $\boldsymbol{X} = f(\boldsymbol{V}) = f \circ f_{\mathrm{RF}}(\boldsymbol{U})$ [108]. This gives rise to a "spurious ICA solution" $(f_{\mathrm{RF}}^{-1} \circ f^{-1}, G_{\mathrm{ICA}})$ where $G_{\mathrm{ICA}}$ denotes the empty graph with independent components. Due to the non-identifiability of nonlinear ICA [54, 83], however, we cannot even learn the composition $f \circ f_{\mathrm{RF}}$, let alone separate it into its constituents $f$ and $f_{\mathrm{RF}}$ to isolate the intermediate causal variables $\boldsymbol{V}$.

Motivated by these challenges to identifiability from i.i.d. data, we instead consider learning from *multiple environments* $e \in \mathcal{E}$. That is, we assume access to heterogenous data from multiple distinct distributions $P_{\boldsymbol{X}}^e$. Environments can arise, for example, from different experimental settings or

---

[1] A diffeomorphism is a bijective function $f$ such that both $f$ and $f^{-1}$ are continuously differentiable.

[2] If they were not in such order to begin with, we could apply an appropriate permutation $\tilde{\pi}$ to $\boldsymbol{V}$ and incorporate the inverse permutation into the unknown mixing $f$ without affecting $\boldsymbol{X} = f(\boldsymbol{V}) = (f \circ \tilde{\pi}^{-1})(\tilde{\pi}(\boldsymbol{V}))$ [117].

[3] $\sim_{\mathrm{CRL}}$ satisfies symmetry and transitivity because permutations and element-wise functions commute.

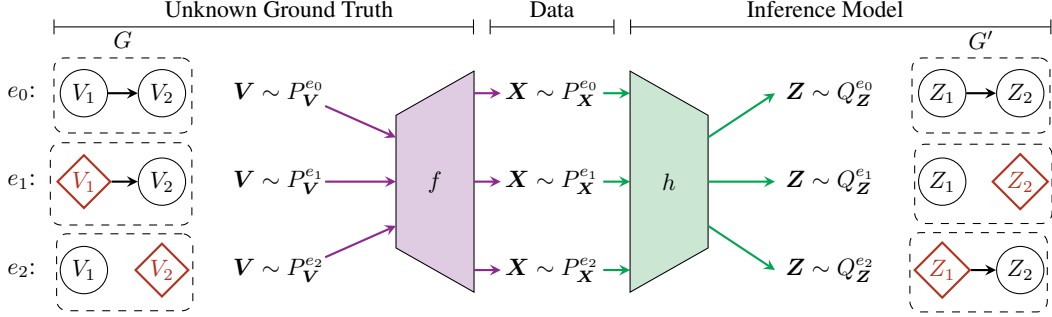

Figure 2: **Multi-Environment Setup with Single-Node Perfect Interventions and Shared Mixing Function.** Illustration of the considered multi-environment setup for $n = 2$ causal variables $\boldsymbol{V} = \{V_1, V_2\}$ with graph $G$ given by $V_1 \rightarrow V_2$, shared mixing function $f$, and environments $\mathcal{E} = \{e_0, e_1, e_2\}$, corresponding to the observational setting ($e_0$) and perfect stochastic interventions on $V_1$ (in $e_1$) and $V_2$ (in $e_2$). The learnt unmixing function, or encoder, is denoted by $h$, and the inferred latent representation by $\boldsymbol{Z} = h(\boldsymbol{X})$. The corresponding inferred latent distributions $Q_{\boldsymbol{Z}}^{e_i} = h_*(P_{\boldsymbol{X}}^{e_i})$ are Markovian w.r.t. the candidate graph $G'$ (here, equal to $G$). Since the intervention targets are not known, they may in principle differ in $Q_{\boldsymbol{Z}}^e$ as shown here. However, as we prove in Thm. 3.2, such misalignment is only possible if a certain genericity condition (3.2) is violated.

correspond to broader contexts such as climate or time. Previous work has shown that this setting can, in principle, provide useful causal learning signals [5, 8, 19, 32, 34, 47, 50, 59, 71–73, 82, 89, 98, 99, 102, 104, 109, 117, 120, 122]. However, multi-environment data is not necessarily useful if the corresponding distributions $P_{\boldsymbol{X}}^e$ are allowed to differ in arbitrary ways. What makes this setting interesting is the assumption that certain parts of the causal generative process are shared across environments.

Here, we assume that all environments share the same invariant mixing function and underlying SCM, and that any distribution shifts arise from interventions on some of the causal mechanisms.[4] General interventions can be modelled in SCMs by replacing some of the equations in (2.1) by new assignments $V_i := \tilde{f}_i(\boldsymbol{V}_{\text{pa}(i)}, \tilde{U}_i)$, resulting in "intervened" mechanisms $\tilde{p}_i(v_i \mid \boldsymbol{v}_{\text{pa}(i)})$ replacing the corresponding conditionals $p_i(v_i \mid \boldsymbol{v}_{\text{pa}(i)})$ in (2.2) [29, 36, 95]. We summarise this as follows.

**Assumption 2.8** (Shared mixing and mechanisms). Each environment $e$ shares the same mixing $f$,

$$P_{\boldsymbol{X}}^e = f_*(P_{\boldsymbol{V}}^e)$$

and each $P_{\boldsymbol{V}}^e$ results from the same SCM through an intervention on a subset of mechanisms $\mathcal{I}^e \subseteq [n]$:

$$p^e(\boldsymbol{v}) = p^e(v_1, ..., v_n) = \prod_{i \in \mathcal{I}^e} p_i^e\left(v_i \mid \boldsymbol{v}_{\text{pa}(i)}\right) \prod_{j \in [n] \setminus \mathcal{I}^e} p_j\left(v_j \mid \boldsymbol{v}_{\text{pa}(j)}\right) .$$

Importantly, the intervention targets $\mathcal{I}^e$ are not assumed to be known.

An intervention on some $V_i$ that fully removes the influence from its parents $\boldsymbol{V}_{\text{pa}(i)}$ is referred to as *perfect*, otherwise as *imperfect*. It has been shown that imperfect interventions are generally insufficient for full identifiability [18], even in the linear case [117]. This is intuitive: if arbitrary imperfect interventions were allowed, including ones which preserve $f_i(\boldsymbol{V}_{\text{pa}(i)}, \cdot)$ and only replace $\tilde{U}_i$ with some new $\tilde{U}_i$, then the spurious ICA solution $(f_{\text{RF}}^{-1} \circ f^{-1}, G_{\text{ICA}})$ should be indistinguishable from the ground truth. In line with prior work [5, 18, 117, 122], we therefore assume perfect interventions.

**Assumption 2.9** (Perfect interventions). For all $e \in \mathcal{E}$ and $i \in \mathcal{I}^e$, we have $p^e(v_i \mid \boldsymbol{v}_{\text{pa}(i)}) = p^e(v_i)$.

Based on the principle of independent causal mechanisms [100, 109], the *sparse mechanism shift hypothesis* [98, 110] posits that distribution changes tend to manifest themselves in a sparse or local way in the causal factorization. In this spirit, we will assume single-node ("atomic") interventions, $|\mathcal{I}^e| = 1$, for our main results, as also required for existing results [5, 18, 117, 122].

As motivated in § 2.2, given data from $\{P_{\boldsymbol{X}}^e\}_{e \in \mathcal{E}}$ we consider candidate solutions of the form $(h, G')$ where $h$ is an unmixing function, or encoder, which maps observations $\boldsymbol{X}$ to the inferred latents $\boldsymbol{Z} = h(\boldsymbol{X})$, and $G'$ is causal graph capturing the relations among the $Z_i$. The corresponding distributions of the inferred latents are thus given by the push-forward

$$Q_{\boldsymbol{Z}}^e = h_*(P_{\boldsymbol{X}}^e) = (h \circ f)_* P_{\boldsymbol{V}}^e$$

The multi-environment setup with unknown single-node perfect interventions is illustrated in Fig. 2.

---

[4]One could also say that there exists only one environment and each dataset corresponds to an intervention.

# 3 Identifiability Theory

We start by showing that identifiability up to $\sim_{\mathrm{CRL}}$ is, in fact, the best we can hope for when learning from interventional data, without any more direct forms of supervision. To this end, we state the following result, which is presented more formally and proven in Appx. B.

**Proposition 3.1** (Minimality of $\sim_{\mathrm{CRL}}$; informal)**.** *Let $\boldsymbol{Z}$ be any representation that is $\sim_{\mathrm{CRL}}$ equivalent to $\boldsymbol{V}$, with $G' = \pi(G)$ the associated DAG. Then for any intervention on $\boldsymbol{V}_{\mathcal{I}} \subseteq \boldsymbol{V}$ in $G$, there exists an equally sparse intervention on $\boldsymbol{Z}_{\pi(\mathcal{I})} \subseteq \boldsymbol{Z}$ in $G'$ inducing the same observed distribution on $\boldsymbol{X}$.*

We now present our identifiability results. We first study the most fundamental bivariate case with two latent causal variables $V_1$ and $V_2$. This can loosely be seen as the CRL analogue of the widely studied cause-effect problem ($X \to Y$ or $Y \to X$?) in classical structure learning [88, 100].

**Theorem 3.2** (Bivariate identifiability up to $\sim_{\mathrm{CRL}}$ from one perfect stochastic intervention per node)**.** *Suppose that we have access to multiple environments $\{P_{\boldsymbol{X}}^e\}_{e \in \mathcal{E}}$ generated as described in § 2 under Asms. 2.2, 2.5, 2.8 and 2.9 with $n = 2$. Let $(h, G')$ be any candidate solution such that the inferred latent distributions $Q_{\boldsymbol{Z}}^e = h_*(P_{\boldsymbol{X}}^e)$ of $\boldsymbol{Z} = h(\boldsymbol{X})$ and the inferred mixing function $h^{-1}$ satisfy the above assumptions w.r.t. the candidate causal graph $G'$. Assume additionally that*

(A1) *all densities $p^e$ and $q^e$ are continuously differentiable and fully supported on $\mathbb{R}^n$;*

(A2) *we have access to a* known observational environment $e_0$ *and one* single node perfect intervention for each node, *with* unknown targets*: there exist $n+1$ environments $\mathcal{E} = \{e_i\}_{i=0}^n$ such that $\mathcal{I}^{e_0} = \varnothing$ and for each $i \in [n]$ we have $\mathcal{I}^{e_i} = \{\pi(i)\}$ for an unknown permutation $\pi$ of $[n]$;*

(A3) *for all $i \in [n]$, the intervened mechanisms $\tilde{p}_i(v_i)$ differ from the corresponding base mechanisms $p_i(v_i \mid \boldsymbol{v}_{\mathrm{pa}(i)})$ everywhere, in the sense that*

$$\forall \boldsymbol{v}: \qquad \frac{\partial}{\partial v_i} \frac{\tilde{p}_i(v_i)}{p_i(v_i \mid \boldsymbol{v}_{\mathrm{pa}(i)})} \neq 0; \tag{3.1}$$

(A4) *(**"genericity"**) the base and intervened mechanisms are not fine-tuned to each other, in the sense that there exists a continuous function $\varphi: \mathbb{R}^+ \to \mathbb{R}$ for which*

$$\mathbb{E}_{\boldsymbol{v} \sim P_{\boldsymbol{V}}^{e_0}} \left[ \varphi\left( \frac{\tilde{p}_2(v_2)}{p_2(v_2 \mid v_1)} \right) \right] \neq \mathbb{E}_{\boldsymbol{v} \sim P_{\boldsymbol{V}}^{e_1}} \left[ \varphi\left( \frac{\tilde{p}_2(v_2)}{p_2(v_2 \mid v_1)} \right) \right] \tag{3.2}$$

*Then the ground truth is identified in the sense of Defn. 2.6, that is, $(f^{-1}, G) \sim_{\mathrm{CRL}} (h, G')$.*

*Proof sketch (full proof in Appx. C.2).* Consider $V_1 \to V_2$ (the proof for $V_1 \not\to V_2$ is similar). Let $\psi = f^{-1} \circ h^{-1}: \mathbb{R}^n \to \mathbb{R}^n$ such that $\boldsymbol{V} = \psi(\boldsymbol{Z})$.[5] By the change of variable formula, for all $\boldsymbol{z}$

$$q^e(\boldsymbol{z}) = p^e(\psi(\boldsymbol{z})) \left| \det \boldsymbol{J}_\psi(\boldsymbol{z}) \right| \tag{3.3}$$

where $(\boldsymbol{J}_\psi(\boldsymbol{z}))_{ij} = \frac{\partial \psi_i}{\partial z_j}(\boldsymbol{z})$ denotes the Jacobian of $\psi$. We consider two separate cases, depending on whether the intervention targets in $q^{e_i}$ for $e_i \in \{e_1, e_2\}$ match those in $p^{e_i}$ (Case 1) or not (Case 2).

*Case 1: Aligned Intervention Targets.* According to Asm. 2.8 and (A2), (3.3) applied to the known observational environment $e_0$ and the interventional environments $e_1, e_2$ leads to the system of equations:

$$q_1(z_1) q_2(z_2 \mid z_{\mathrm{pa}(2;G')}) = p_1\left(\psi_1(\boldsymbol{z})\right) p_2\left(\psi_2(\boldsymbol{z}) \mid \psi_1(\boldsymbol{z})\right) \left| \det \boldsymbol{J}_\psi(\boldsymbol{z}) \right| \tag{3.4}$$

$$\tilde{q}_1(z_1) q_2(z_2 \mid z_{\mathrm{pa}(2;G')}) = \tilde{p}_1\left(\psi_1(\boldsymbol{z})\right) p_2\left(\psi_2(\boldsymbol{z}) \mid \psi_1(\boldsymbol{z})\right) \left| \det \boldsymbol{J}_\psi(\boldsymbol{z}) \right| \tag{3.5}$$

$$q_1(z_1) \tilde{q}_2(z_2) = p_1\left(\psi_1(\boldsymbol{z})\right) \tilde{p}_2\left(\psi_2(\boldsymbol{z})\right) \left| \det \boldsymbol{J}_\psi(\boldsymbol{z}) \right| \tag{3.6}$$

where $z_{\mathrm{pa}(2;G')}$ denotes the parents of $z_2$ in $G'$. Taking quotients of (3.5) and (3.4) yields

$$\frac{\tilde{q}_1}{q_1}(z_1) = \frac{\tilde{p}_1}{p_1}(\psi_1(\boldsymbol{z})) \quad \overset{\frac{\partial}{\partial z_2}}{\implies} \quad 0 = \left( \frac{\tilde{p}_1}{p_1} \right)'(\psi_1(\boldsymbol{z})) \frac{\partial \psi_1}{\partial z_2}(\boldsymbol{z}) \quad \overset{(A3)}{\implies} \quad \frac{\partial \psi_1}{\partial z_2}(\boldsymbol{z}) = 0. \tag{3.7}$$

Thus $V_1 = \psi_1(Z_1)$ and $q_1(z_1) = p_1(\psi_1(z_1)) \left| \frac{\partial \psi_1}{\partial z_1}(z_1) \right|$. Substitution into (3.6) yields

$$\tilde{q}_2(z_2) = \tilde{p}_2\left(\psi_2(z_1, z_2)\right) \left| \frac{\partial \psi_2}{\partial z_2}(z_1, z_2) \right| \tag{3.8}$$

---

[5]By Asm. 2.5, $f$, $h$, and thus also $h \circ f$ are diffeomorphisms. Hence, $\psi$ is well-defined and also diffeomorphic.

where we have used that, according to (3.7), $\boldsymbol{J}_\psi$ is triangular. According to (3.8), for all $z_1$, the mapping $z_2 \mapsto \psi_2(z_1, z_2)$ is measure preserving for $\tilde{q}_2$ and $\tilde{p}_2$. By Lemma C.1 [18, § A.2, Lemma 2], it follows that $\psi_2$ must be constant in $z_1$.[6] Hence, $\psi$ is an element-wise function. Finally, $G = G'$ follows from faithfulness (Asm. 2.2), for otherwise $(V_1, V_2) = (\psi_1(Z_1), \psi_2(Z_2))$ would be independent.

*Case 2: Misaligned Intervention Targets.* If $G' \neq G$, a similar argument to Case 1 (with the roles of $z_1$ and $z_2$ interchanged) also yields a contradiction to faithfulness. This leaves $G = G'$. Writing down the system of equations similar to (3.4)–(3.6), and then taking ratios of $e_1$ and $e_2$ with $e_0$ yields

$$\frac{\tilde{q}_1}{q_1}(z_1) = \frac{\tilde{p}_2(\psi_2(\boldsymbol{z}))}{p_2(\psi_2(\boldsymbol{z}) \mid \psi_1(\boldsymbol{z}))} \qquad \text{and} \qquad \frac{\tilde{q}_2(z_2)}{q_2(z_2 \mid z_1)} = \frac{\tilde{p}_1}{p_1}(\psi_1(\boldsymbol{z})) . \tag{3.9}$$

These conditions highlight the misalignment in intervention targets (see Fig. 2). Unlike in Case 1, they do not directly imply that some elements of $\boldsymbol{J}_\psi$ need to be zero, that is $\boldsymbol{Z}$ can be arbitrarily mixed w.r.t. $\boldsymbol{V}$. However, (3.9) imposes constraints on the form of $\psi$ that, by exploiting the invariance of $q_1$ across $e_0$ and $e_1$ while $p_1$ changes to $\tilde{p}_1$, can ultimately be shown to only be satisfied if the two expectations in (3.2) are equal for all continuous $\varphi$. However, such fine-tuning is ruled out by (A4). $\square$

*Remark 3.3.* The main difficulty of the proof is that (3.9) may, in principle, hold when $(p, \tilde{p}, q, \tilde{q})$ and $\psi$ are completely unconstrained. This does not arise in prior work if the intervention targets are known [80, 81, 126] (Case 1), or the densities or mixing are parametrically constrained [3, 117, 122].

**On the Genericity Condition (A4).** The condition in (3.2) contrasts expectations of the same quantity w.r.t. the observational distribution $P_{\boldsymbol{V}}^{e_0}$ and the interventional distribution $P_{\boldsymbol{V}}^{e_1}$ corresponding to an intervention on $V_1$ that turns $p_1$ into $\tilde{p}_1$. The shared argument, on the other hand, is a function of the ratio between the intervened mechanism $\tilde{p}_2$ and its base mechanism $p_2$. While the two expectations are always equal for linear $\varphi$, other choices imply non-trivial constraints. For instance, $\varphi(y) = y^2$ yields

$$\int \left(\tilde{p}_1(v_1) - p_1(v_1)\right) \int \frac{\tilde{p}_2(v_2)^2}{p_2(v_2 \mid v_1)} \, \mathrm{d}v_2 \, \mathrm{d}v_1 \neq 0 .$$

Since $p_1 \neq \tilde{p}_1$ by assumption (A3), we argue that (A4) should generally hold for randomly chosen $(p_1, p_2, \tilde{p}_1, \tilde{p}_2)$ and can only be violated if they are fine-tuned to each other. It can thus be viewed as encoding some notion of genericity—in line with the principle of independent causal mechanisms [15, 43, 61, 100, 109], but also involving the intervened mechanisms. Interestingly, related genericity conditions also arise in the study of nonlinear cause-effect inference from observational data, where identifiability is often obtained up to a set of pathological ("fine-tuned") spurious solutions satisfying a partial differential equation involving the original mechanisms [49, 58, 118, 137]. Further, we note that $\varphi$ in (3.2) can also be thought of as a *witness function of genericity*, similar to witness functions in kernel-based two sample and independence testing [44].

Next, we provide our identifiability result for an arbitrary number of causal variables.

**Theorem 3.4** (Identifiability up to $\sim_{\mathrm{CRL}}$ from two paired perfect stochastic interventions per node)**.** *Suppose that we have access to multiple environments $\{P_{\boldsymbol{X}}^e\}_{e \in \mathcal{E}}$ generated as described in § 2 under Asms. 2.2, 2.3, 2.5, 2.8 and 2.9. Let $(h, G')$ be any candidate solution such that the inferred latent distributions $Q_{\boldsymbol{Z}}^e = h_*(P_{\boldsymbol{X}}^e)$ of $\boldsymbol{Z} = h(\boldsymbol{X})$ and the inferred mixing function $h^{-1}$ satisfy the above assumptions w.r.t. the candidate causal graph $G'$. Assume additionally that*

*(A1) all densities $p^e$ and $q^e$ are continuously differentiable and fully supported on $\mathbb{R}^n$;*

*(A2') we have access to at least one pair of single-node perfect interventions per node, with unknown targets: there exist $m \geq n$ known pairs of environments $\mathcal{E} = \{(e_j, e_j')\}_{j=1}^m$ such that for each $i \in [n]$ there exists some unknown $j \in [m]$ for which $\mathcal{I}^{e_j} = \mathcal{I}^{e_j'} = \{i\}$;*

*(A3') for all $i \in [n]$, the intervened mechanisms $\tilde{p}_i(v_i)$ and $\tilde{\tilde{p}}_i(v_i)$ differ everywhere, in the sense that*

$$\forall v_i : \qquad \left(\frac{\tilde{\tilde{p}}_i}{\tilde{p}_i}\right)'(v_i) \neq 0 ; \tag{3.10}$$

*Then the ground truth is identified in the sense of Defn. 2.6, that is, $(f^{-1}, G) \sim_{\mathrm{CRL}} (h, G')$.*

---

[6]This step is where the assumption of perfect interventions (Asm. 2.9) is leveraged: the conclusion would not hold for arbitrary imperfect interventions for which (3.8) would involve $\tilde{q}_2(z_2 \mid z_1)$ and $p_2(\psi_2(z_1, z_2) \mid \psi_1(z_1))$.

*Proof sketch (full proof in Appx. C.3).* By considering ratios between $e_j$ and $e'_j$, taking partial derivatives w.r.t. $z_l$, and using assumptions (A3'), we can identify a subset $\mathcal{E}_n \subseteq \mathcal{E}$ of exactly $n$ pairs of environments corresponding to distinct intervention targets in $p$ (for otherwise $\psi$ cannot be invertible). For $(e_i, e'_i) \in \mathcal{E}_n$, w.l.o.g. fix the intervention targets in $p$ to $\mathcal{I}^{e_i} = \mathcal{I}^{e'_i} = \{i\}$ and let $\pi$ be a permutation of $[n]$ such that $\pi(i)$ denotes the inferred intervention target in $q$ that by (A2') is shared across $(e_i, e'_i)$. By the same argument as before, we must have $V_i = \psi_i(Z_{\pi(i)})$, that is, $\psi$ is a permutation composed with an element-wise function. It remains to show that $\pi$ is a graph isomorphism, that is, $V_i \to V_j$ in $G \iff Z_{\pi(i)} \to Z_{\pi(j)}$ in $G'$. We prove $\implies$; the other direction is analogous. Suppose for a contradiction that there exist $(i, j)$ such that $V_i \to V_j$ in $G$, but $Z_{\pi(i)} \not\to Z_{\pi(j)}$ in $G'$. Consider $e_i$ in which there are perfect interventions on $Z_{\pi(i)}$ and $V_i$. For $Z_{\pi(k)} \in \mathbf{Z}_{\mathrm{pa}(\pi(j);G')}$, let $\tilde{V}_k = \psi_k(Z_{\pi(k)})$ and denote $\tilde{\mathbf{V}} = \cup_k \tilde{V}_k \subset \mathbf{V} \setminus \{V_i, V_j\}$. Since $Z_{\pi(i)}$ and $Z_{\pi(j)}$ are d-separated by $\mathbf{Z}_{\mathrm{pa}(\pi(j);G')}$ in the post-intervention graph $G'_{\overline{Z}_{\pi(i)}}$ with arrows pointing into $Z_{\pi(i)}$ removed [95], it follows by Markovianity of $q$ w.r.t. $G'$ that $Z_{\pi(i)} \perp\!\!\!\perp Z_{\pi(j)} \mid \mathbf{Z}_{\mathrm{pa}(\pi(j);G')}$ in $Q_{\mathbf{Z}}^{e_i}$. By applying the corresponding diffeomorphic functions $\psi_i$, it follows from Lemma C.2 in Appx. C.1 that $V_i \perp\!\!\!\perp V_j \mid \tilde{\mathbf{V}}$ in $P_{\mathbf{V}}^{e_i}$. This violates faithfulness (Asm. 2.2) of $P_{\mathbf{V}}$ to $G$ since $V_i$ and $V_j$ are d-connected in $G_{\overline{V}_i}$. $\square$

(A2') states that we know that a *pair of datasets* corresponds to two distinct interventions on the same underlying variable, even though we may not know the exact target of the intervention. This situation could arise, for example, if both datasets are collected under the same experimental setup but with varying experimental parameters. We stress that this is different from data consisting of *pairs of views* $(\mathbf{x}, \tilde{\mathbf{x}})$ sharing the values of some variables, which is *counterfactual* in nature [4, 18, 123]. One of the main challenges for our analysis (compared to a counterfactual multi-view setting) thus stems from the lack of correspondences across observations from different datasets. We also stress that a purely observational environment is not needed in this case, cf. (A2) in Thm. 3.2.

(A3') states that the intervention mechanisms are distinct in that their ratio is strictly monotonic, similar to (3.1) in (A3). This is a slightly stronger version of the assumption of *interventional discrepancy* proposed by Wendong et al. [126],[7] which has been shown to be necessary for identifiability even if the graph $G$ is known. For Gaussian $\tilde{p}_i, \tilde{\tilde{p}}_i$, (A3') is satisfied, for example, by a shift in mean. In the proofs, this is used to show that $\psi$ must be an element-wise function, see (3.7). Intuitively, if $\tilde{p}_i = \tilde{\tilde{p}}_i$ in some open set for more than one $i$, then the underlying variables can be nonlinearly mixed by a measure-preserving automorphism within this set without affecting the observed distributions [126].

## 4 Interpreting Causal Representations

Suppose that we succeed in identifying $\mathbf{V}$ and $G$ up to $\sim_{\mathrm{CRL}}$ (Defn. 2.6). How can we use or interpret such a causal representation? Since the scale of the variables is arbitrary (§ 2.2), we clearly cannot predict the exact outcomes of interventions. We therefore seek causal quantities that are preserved by the irresolvable ambiguities of $\sim_{\mathrm{CRL}}$. A prime candidate for this are interventional causal notions defined in terms of information theoretic quantities [28] and in particular the KL divergence $D_{\mathrm{KL}}$.

**Definition 4.1** (Causal influence; based on Defn. 2 of Janzing et al. [62])**.** Let $P_{\mathbf{V}}$ be Markovian w.r.t. a DAG $G$ with vertices $\mathbf{V}$. For any $V_i \to V_j$ in $G$, the *causal influence of $V_i$ on $V_j$* is given by

$$\mathfrak{C}_{i \to j}^{P_{\mathbf{V}}} := D_{\mathrm{KL}}\big(P_{\mathbf{V}} \,\|\, P_{\mathbf{V}}^{i \to j}\big), \quad \text{where} \quad p_j^{i \to j}\big(v_j \mid \mathbf{v}_{\mathrm{pa}(j) \setminus \{i\}}\big) = \int p_j\big(v_j \mid \mathbf{v}_{\mathrm{pa}(j)}\big) p_i(v_i) \, \mathrm{d}v_i$$

and $P_{\mathbf{V}}^{i \to j}$ is the interventional distribution arising from replacing the $j^{\mathrm{th}}$ mechanism by $p_j^{i \to j}$.[8]

The following result, proven in Appx. C.4, states that the causal influences are invariant to reparametrisation and equivariant to permutations, the two irresolvable ambiguities of the $\sim_{\mathrm{CRL}}$ equivalence class.

**Theorem 4.2** (Preservation of causal influences under $\sim_{\mathrm{CRL}}$)**.** *Let $P_{\mathbf{V}}$ be Markovian w.r.t. $G$, let $\pi$ be a graph isomorphism of $G$, and let $\phi$ be an element-wise diffeomorphism. Let $\mathbf{Z} = \mathbf{P}_{\pi^{-1}} \circ \phi(\mathbf{V})$ and denote its induced distribution by $Q_{\mathbf{Z}}$. Then for any $V_i \to V_j$ in $G$ we have $\mathfrak{C}_{i \to j}^{P_{\mathbf{V}}} = \mathfrak{C}_{\pi(i) \to \pi(j)}^{Q_{\mathbf{Z}}}$.*

Thm. 4.2 implies that the strength of causal relations among variables in the inferred graph carry meaning. They can thus be used to uncover changes to the latent causal mechanisms underlying different experimental datasets, for example, to gain scientific insights when combined with domain knowledge.

---

[7]Cf. the *interventional regularity* assumption of Varici et al. [122, Asm. 2] which instead considers partial derivatives w.r.t. the parents and is related to c-faithfulness [59, Defn. 7].

[8]Intuitively, this intervention captures the process of removing the arrow $V_i \to V_j$ in $G$ and "feeding" the conditional $p(v_j \mid \mathbf{v}_{\mathrm{pa}(j)})$ with an independent copy of $V_i$, distributed according to its marginal, see [62] for details.

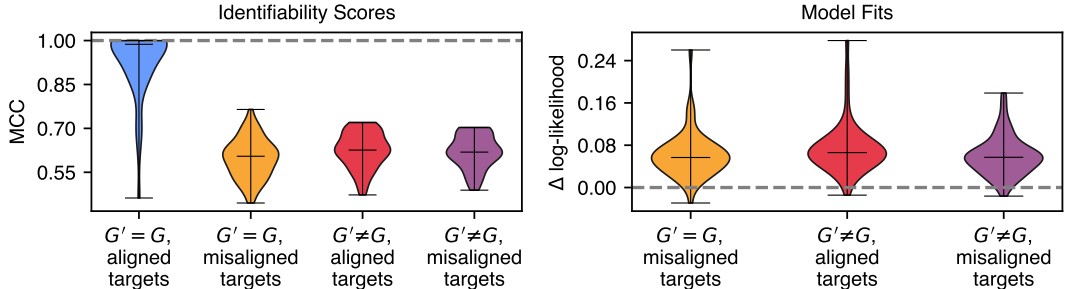

**Figure 3: Empirical Comparison of Correctly and Incorrectly Specified Normalizing Flow-Based Models.**
For $n = 2$ latent causal variables with graph $G$ given by $V_1 \to V_2$, we compare a generative model based on the correct causal graph $G' = G$ and intervention targets (blue) to other generative models assuming the wrong graph $G' \neq G$ or misaligned intervention targets (yellow, red, purple). We show mean correlation coefficients (MCCs) between the learned and ground truth latents *(Left)* and the difference in validation model log-likelihood between the well-specified and misspecified models *(Right)*. Each violin plot is based on 50 different ground truth data generating processes; the horizontal lines indicate the minimum, median and maximum values.

## 5 Learning Objectives

While our main focus is on studying identifiability, our theoretical insights also suggest approaches to learning causal representations from finite interventional datasets sampled from $\{P_X^e\}_{e \in \mathcal{E}}$. The main idea is to fit the data in a way that preserves the sparsity of interventions [98, 110] by employing the same (un)mixing function and sharing mechanisms across environments (Asm. 2.8). We sketch two approaches, which, according to our theory, should both asymptotically identify the ground truth up to $\sim_{\mathrm{CRL}}$ if the set of available environments $\mathcal{E}$ is sufficiently diverse (and the other assumptions hold).

- *Autoencoder Framework*: Jointly learn an encoder $h$, a graph $G'$, and intervention targets $\mathcal{I}^e$ such that the encoded latents $\boldsymbol{Z} = h(\boldsymbol{X})$ can be used to reconstruct the observed $\boldsymbol{X}$ across all environments, while ensuring all but the intervened mechanisms are shared. Using $E$ as an environment indicator, the latter corresponds to the constraint $Z_i \perp\!\!\!\perp E \mid \boldsymbol{Z}_{\mathrm{pa}(i;G')}$ for $i \notin \mathcal{I}^e$, implementable, for example, through a suitable conditional independence test [39, 94, 138].

- *Generative Modelling Approach*: Fit a base generative model $(G', p, f)$ and intervention models $(\mathcal{I}^e, \{p_i^e\}_{i \in \mathcal{I}^e})_{e \in \mathcal{E}}$ by maximizing the likelihood of the multi-environment data. For example, given a candidate graph $G'$ and candidate intervention targets $\{\mathcal{I}^e\}_{e \in \mathcal{E}}$, learn the base and intervened mechanisms and mixing; then pick the graph and intervention targets that achieve the best fit.

## 6 Experiments

**Setup.** We experimentally pursue the second, generative approach. Specifically, we model the mixing function generating $\boldsymbol{X}$ from $\boldsymbol{Z}$ as a *normalizing flow* [30, 93] with different environment-specific base distributions, determined by the underlying causal graph, intervention targets, and (learnt) base and intervened mechanisms. Here, we focus on the bivariate case with two ground-truth causal latent variables $V_1 \to V_2$. According to Thm. 3.2, this setting should be identifiable from three environments: an observational one and one perfect intervention on each of $V_1$ and $V_2$. Our goal is to verify this empirically in light of finite data and optimization issues. We fix the observation dimension to $d = 2$ to facilitate exact likelihood training of the normalizing flows, and fit a separate generative model for each choice of graph and intervention targets.[9] The base and intervened mechanisms are linear Gaussian and the mixing function a three-layer MLP, see Appx. D.1 for implementation details.

**Results.** The results are summarized in Fig. 3. Our main findings are two-fold: First, we observe that, in the majority of cases, the well-specified model attains the highest held-out log-likelihood, as shown in Fig. 3 *(Right)*. This suggests that the likelihood of otherwise comparable generative models can act as a reliable criterion to select the correct causal graph and intervention targets. Second, we find that the ground truth latent causal variables are approximately identified up to element-wise rescaling (MCC values close to one) by the correctly specified model and not by any other model, as shown in Fig. 3 *(Left)*. This indicates that recovering the correct graph and targets is not only sufficient but also necessary for reliable identification of the causal representation. Taken together, these findings are consistent with our notion of $\sim_{\mathrm{CRL}}$-identifiability (Defn. 2.6) and Thm. 3.2.

---

[9]This can be viewed as a nested approach in which the inner loop corresponds to the method for Causal Component Analysis (CauCA) of Wendong et al. [126], and the outer loop to a search over $G'$ and $\{\mathcal{I}^e\}_{e \in \mathcal{E}}$. Code to reproduce our experiments is available at: https://github.com/akekic/causal-component-analysis.

In Appx. D.2, we perform an additional experiment with $n = 3$ variables, nonlinear latent SCMs, fixed causal order without graph search, and one interventional environment per node, thus assaying violations of assumption (A2') in the context of Thm. 3.4. Generally, we find that a correct choice of intervention targets can be selected based on model likelihood, leading to approximate identification of the causal variables, even when only the causal order is given, see Fig. 4 and Appx. D.2 for details.

# 7 Conclusions and Discussion

The world is full of domain shifts and different environments. Often, we cannot pin down what exactly differs between two domains, but it may reasonably be modelled as a change in some causal mechanisms. While prior work relied on harder-to-obtain counterfactual data or parametric constraints for identifiability, our study demonstrates that interventional data can be sufficient—even in the nonparametric case where the spaces of mixing functions and mechanisms are infinite dimensional. Our results can be considered a step towards justifying the use of expressive machine learning methods for learning interpretable causal representations from high-dimensional experimental data in situations where parametric assumptions are undesirable, e.g., for complex systems in physics, biology, or medicine.

**Weaker Notions of Identifiability.** In this work, we have focused on the strongest notion of identifiability that is achievable in a nonparametric setting (Defn. 2.6). However, subject to the available data and assumptions, identifiability up to $\sim_{\mathrm{CRL}}$ is not always possible. In this case, weaker notions of identifiability are of interest. For example, we may not be able to uniquely recover variables that are not targeted by interventions [5, 80, 123], or only recover groups of variables up to (non-)linear mixing [5, 122, 123] and the graph up to transitive closure if interventions are imperfect, or soft [117, 136]. A precise characterization of weaker notions of *nonparametric* identifiability from different types of *interventions* (cf. [74] for a temporal, semi-parametric setting) is an interesting direction for future work.

**Known vs. Unknown Intervention Targets.** When intervention targets can be considered known appears to be a more nuanced concept in CRL than in a fully observed setting, see also [126, §E] for an extended discussion. Recall that we assume w.l.o.g. that $V_1 \preceq ... \preceq V_n$ and only consider graphs respecting this ordering (§ 2.2), see also [117, Remark 1]. The intervention targets are then unknown w.r.t. the pre-imposed causal ordering. This is a key aspect that makes our setting more realistic, but also substantially complicates the analysis (see (3.9) and Remark 3.3). Similar to [122], for Thm. 3.2 we require a set of *exactly* $n$ environments (one intervention on each node).[10] However, we relax this requirement to mere coverage ("at least 1") in Thm. 3.4 as shown for linear causal models in [22, 117].

**Identifiability From One Intervention per Node for Any $n$.** We conjecture that Thm. 3.2 can be generalized to $n > 2$, subject to a suitably adjusted set of genericity conditions involving several intervened and base mechanisms, akin to (3.2) in the bivariate case. The main challenge to such a generalization appears to be combinatorial, as there are $n! - 1$ ways of misaligning intervention targets across $p$ and $q$. In Thm. 3.4, we sidestep this issue by assuming pairs of environments. Thus, while two single-node perfect interventions are sufficient, we do not believe this to be necessary.

**On the Assumption of Known $n$ and Its Relation to Markovianity.** Asm. 2.3 relates to *causal sufficiency* or *Markovianity* in classical causal inference, which correspond to the assumption of independent $U_i$ in Defn. 2.1, implying the causal Markov factorization (2.2) [95, 100, 115]. With *unobserved* $V$ and *unknown* $n$, the notion of "unobserved confounders" gets blurred, since one can, in principle, always construct a causally sufficient system by increasing $n$ and adding any causes of two or more endogenous variables to $V$. Asm. 2.3 then states that the minimum number of variables required to do so is known.[11] However, this can lead to very large systems, which may in turn challenge the assumption of an invertible mixing (Asm. 2.5). Extensions of our analysis to unknown $n$ [63, 68], non-Markovian, or non-invertible [131] models constitute an interesting direction for future investigations.

**Practicality.** The method explored in § 6 requires searching over graphs and intervention targets, which gets intractable even for moderate $n$. Simultaneously learning an (un)mixing function, causal graph, intervention targets, and mechanisms is challenging. Further work is needed to make methods for nonparametric CRL from multi-environment data more practical, e.g., by exploring the proposed autoencoder framework with a continuous parametrisation of graph [139, 140] and targets [60].

---

[10]By dropping the assumption of a fixed ordering and considering all DAGs, in this case one could also call $V_i$ the variable intervened upon in environment $i$, and then consider the targets "known" in this sense.

[11]Techniques for estimating the intrinsic dimensionality $n$ of the observation manifold $\mathcal{X} \subseteq \mathbb{R}^d$ or methods rooted in Bayesian nonparametrics could provide a means of relaxing this assumption.

## Acknowledgments and Disclosure of Funding

We thank Simon Buchholz, Jiaqi Zhang, and the anonymous reviewers for helpful discussions, comments, and suggestions.

This work was supported by the Tübingen AI Center and by the Deutsche Forschungsgemeinschaft (DFG, German Research Foundation) under Germany's Excellence Strategy, EXC number 2064/1, Project number 390727645. L.G. was supported by the VideoPredict project, FKZ: 01IS21088.

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

# Appendices

## Table of Contents

# A Extended Discussion of Related Work

Prior work on causal representation learning with general nonlinear relationships (both among latents and between latents and observations) and without an explicit task or label typically relies on some form of *weak supervision*. One example of weak supervision is "multi-view" data consisting of tuples of related observations. von Kügelgen et al. [123] consider counterfactual pairs of observations arising from imperfect interventions through data augmentation, and prove identifiability for the invariant non-descendants of intervened-upon variables. Brehmer et al. [18] also use counterfactual pre- and post-intervention views and show that the latent SCM can be identified given all single-node perfect stochastic interventions. Another type of weak-supervision is temporal structure [2], possibly combined with nonstationarity [132, 133], interventions on known targets [80, 81], or observed actions inducing sparse mechanism shifts [74, 75, 110]. Other works use more explicit supervision in the form of annotations of the ground truth causal variables or a known causal graph [111, 126, 130].

A different line of work instead approaches causal representation learning from the perspective of causal discovery in the presence of latent variables [115]. Doing so from *purely observational i.i.d. data* requires additional constraints on the generative process, such as restrictions on the graph structure or particular parametric and distributional assumptions, and typically leverages the identifiability of linear ICA [27, 38, 79]. For *linear, non-Gaussian* models, Silva et al. [113] show that the causal DAG can be recovered up to Markov equivalence if all observed variables are "pure" in that they have a unique latent causal parent. Cai et al. [23] and Xie et al. [128, 129] extend this result to identify the full graph given two pure observed children per latent, and Adams et al. [1] provide sufficient and necessary graphical conditions for full identification. For *discrete* causal variables, Kivva et al. [68] introduce a similar "no twins" condition to reduce the task of learning the number and cardinality of latents and the Markov-equivalence class of the graph to mixture identifiability.

Other lines of work have investigated the relationship between causal models at different levels of coarse-graining or abstraction [6, 11, 12, 24–26, 35, 105, 134, 135], or learning invariant representations in a supervised setting [3, 8, 32, 33, 73, 86, 92, 124, 125], often for domain generalization.

Other concurrent works address, e.g., learning from soft interventions with polynomial mixing [136], or inferring both causal graph and the number of latents subject to graphical constraints [63].

## A.1 Comparison of Related Identifiability Results

To complement the presentation of related multienvironment CRL works in § 1, we provide a structured overview of and comparison with existing identifiability results for causal representation learning in Tab. 1. The table categorizes work along different dimensions. First, we make a distinction based on the type of data (observational, interventional, or counterfactual) results rely on (colour coded in green, yellow, and red, respectively). These are also referred to as different rungs in the "ladder of causation" [97] or layers in the Pearl Causal Hierarchy [10]. Second, we categorize work depending on the assumptions placed on the latent causal model and the mixing function. As can be seen, works relying solely on observational data often require restrictive graphical assumptions on the mixing function. On the other hand, access to much more informative counterfactual data has allowed identification even for nonparametric causal models and mixing functions. Our work can be viewed as a step towards addressing the lack of nonparametric identifiability results in the interventional regime.

We emphasize that Tab. 1 is not exhaustive: certain relevant works did not easily fit into our categorization or the causal representation learning framework adopted in the present work. For example, not listed are works that leverage temporal structure [2, 74, 75, 80, 81], rely on heterogenous data and distribution shifts which are not directly expressed in terms of or linked to interventions [82, 132, 133], allow for edges from observed to latent variables [1], or require more direct supervision [111, 130].

Table 1: **Comparison of Existing Identifiability Results for Causal Reresentation Learning.** All of the listed works assume invertibility (or injectivity) of the mixing function, as well as causal sufficiency (Markovianity) for the causal latent variables. Most or all of the listed results require additional technical assumptions, and may provide additional results, which we omit for sake of readability; see the references for more details.

| Work | Layer | Causal Model | Mixing Function | Main Identifiability Result |
|---|---|---|---|---|
| Cai et al. [23], Xie et al. [128, 129] | observational | linear, non-Gaussian | linear with non-Gaussian noise s.t. each $V_i$ has 2 pure (obs. or unobs.) children | number of latents $+ G$ |
| Kivva et al. [68] | observational | discrete, nonparametric | indentifiable mixture model s.t. obs. children of $V_i \not\subseteq$ obs. children of $V_j$ | number, cardinality & dist. of discrete latents $+ G$ up to Markov equivalence |
| Ahuja et al. [5, Thm. 4] | observational | nonlinear w. independent support [103, 125] | finite-degree polynomial | $V$ up to permutation, shift, & linear scaling |
| Squires et al. [117, Thms. 1 & 2] | interventional | linear | linear | $G$ and $V$ up to partial-order preserving permutations from obs. dist. & all single-node *perfect* interventions |
| Squires et al. [117, Thm. 1] | interventional | linear | linear | $G$ up to transitive closure from obs. dist. & all single-node *imperfect* interventions |
| Varici et al. [122, Thm. 16] | interventional | nonparametric | linear | $G$ and $V$ up to partial-order preserving permutations from obs. dist. & all single-node *perfect* interventions |
| Ahuja et al. [5, Thm. 2] | interventional | nonparametric | finite-degree polynomial | $V$ up to permutation, shift, and linear scaling from all single-node *perfect hard* interventions |
| Buchholz et al. [22] | interventional | linear Gaussian | nonparametric | $G$ and $V$ up to permutation from obs. dist. & all single-node *perfect* interventions |
| **This Work** (Thm. 3.2) | interventional | nonparametric | nonparametric | for $n = 2$: $G$ and $V$ up to $\sim_{\text{CRL}}$ from all single-node *perfect* interventions, subject to genericity (3.2) |
| **This Work** (Thm. 3.4) | interventional | nonparametric | nonparametric | $G$ and $V$ up to $\sim_{\text{CRL}}$ from two distinct, paired single-node *perfect* interventions per node |
| von Kügelgen et al. [123] | counterfactual | nonparametric | nonparametric | block of non-descendants $V_{\text{nd}(\mathcal{I})}$ up to invertible function from fat-hand *imperfect* interventions on $V_{\mathcal{I}}$ |
| Brehmer et al. [18] | counterfactual | nonparametric | nonparametric | $G$ and $V$ up to $\sim_{\text{CRL}}$ from all single-node *perfect* interventions |

# B Proof of Minimality of the CRL Equivalence Class $\sim_{\mathrm{CRL}}$

First, let us recall the main statements from § 2.2.

**Definition 2.6** ($\sim_{\mathrm{CRL}}$-identifiability)**.** Let $\mathcal{H}$ be a space of unmixing functions $h : \mathcal{X} \to \mathbb{R}^n$ and let $\mathcal{G}$ be the space of DAGs over $n$ vertices. Let $\sim_{\mathrm{CRL}}$ be the equivalence relation on $\mathcal{H} \times \mathcal{G}$ defined as

$$(h_1, G_1) \sim_{\mathrm{CRL}} (h_2, G_2) \quad \Longleftrightarrow \quad (h_2, G_2) = (\boldsymbol{P}_{\pi^{-1}} \circ \phi \circ h_1, \pi(G_1)) \tag{2.3}$$

for some element-wise diffeomorphism $\phi(\boldsymbol{v}) = (\phi_1(v_1), \dots, \phi_n(v_n))$ of $\mathbb{R}^n$ and a permutation $\pi$ of $[n]$ such that $\pi : G_1 \mapsto G_2$ is a graph isomorphism and $\boldsymbol{P}_\pi$ the corresponding permutation matrix.

**Proposition 3.1** (Minimality of $\sim_{\mathrm{CRL}}$; informal)**.** *Let $\boldsymbol{Z}$ be any representation that is $\sim_{\mathrm{CRL}}$ equivalent to $\boldsymbol{V}$, with $G' = \pi(G)$ the associated DAG. Then for any intervention on $\boldsymbol{V}_{\mathcal{I}} \subseteq \boldsymbol{V}$ in $G$, there exists an equally sparse intervention on $\boldsymbol{Z}_{\pi(\mathcal{I})} \subseteq \boldsymbol{Z}$ in $G'$ inducing the same observed distribution on $\boldsymbol{X}$.*

We now restate this result more formally.

**Proposition B.1** (Minimality of $\sim_{\mathrm{CRL}}$)**.** *Let $(h, G') \sim_{\mathrm{CRL}} (f^{-1}, G)$ with $\pi$ denoting the graph isomorphism mapping $G$ to $G'$ (i.e., a permutation that preserves the partial topological order of $G$). Let $\boldsymbol{Z} = h(\boldsymbol{X})$ be the inferred representation with distribution $Q_{\boldsymbol{Z}} = h_*(P_{\boldsymbol{X}})$ Markov w.r.t. $G'$ and associated density $q$. Let $\mathcal{I}^e \subseteq [n]$ denote a set of intervention targets, and consider an intervention that changes $p_i(v_i \mid \boldsymbol{v}_{\mathrm{pa}(i)})$ to some intervened mechanism $\tilde{p}_i(v_i \mid \boldsymbol{v}_{\mathrm{pa}(i)})$ for all $i \in \mathcal{I}^e$, giving rise to the interventional distributions $P_{\boldsymbol{V}}^e$ and $P_{\boldsymbol{X}}^e = f_*(P_{\boldsymbol{V}}^e)$. Then there exist appropriately chosen $\tilde{q}_{\pi(i)}(z_{\pi(i)} \mid \boldsymbol{z}_{\mathrm{pa}(\pi(i), G')})$ for $i \in \mathcal{I}^e$ such that the resulting interventional distribution $Q_{\boldsymbol{Z}}^e$ gives rise to the same observed distributions, that is, $P_{\boldsymbol{X}}^e = h_*^{-1}(Q_{\boldsymbol{Z}}^e)$.*

*Proof.* Since $(h, G') \sim_{\mathrm{CRL}} (f^{-1}, G)$, by Defn. 2.6 we have

$$\boldsymbol{Z} = \boldsymbol{P}_{\pi^{-1}} \circ \phi(\boldsymbol{V}) \tag{B.1}$$

for some element-wise diffeomorphism $\phi$ with inverse $\psi = \phi^{-1}$. Then (B.1) implies that for all $i \in [n]$

$$V_i = \psi_i(Z_{\pi(i)}) \tag{B.2}$$

According to (B.2), each conditional in the Markov factorization of $Q_{\boldsymbol{Z}}$ is given in terms of $p$ by

$$q_{\pi(i)}\left(z_{\pi(i)} \mid \boldsymbol{z}_{\mathrm{pa}(\pi(i); G')}\right) = p_i\left(\psi_i\left(z_{\pi(i)}\right) \mid \psi_{\mathrm{pa}(i)}\left(\boldsymbol{z}_{\mathrm{pa}(\pi(i); G')}\right)\right) \left| \frac{\mathrm{d}\psi_i}{\mathrm{d}z_{\pi(i)}}\left(z_{\pi(i)}\right) \right| \tag{B.3}$$

where we have used the change of variables in (B.2) and the fact that $\pi(\mathrm{pa}(i)) = \mathrm{pa}(\pi(i); G')$ since $\pi : G \mapsto G'$ is a graph isomorphism.

Consider an intervention that changes $p_i(v_i \mid \boldsymbol{v}_{\mathrm{pa}(i)})$ to some intervened mechanism $\tilde{p}_i(v_i \mid \boldsymbol{v}_{\mathrm{pa}(i)})$ for all $i \in \mathcal{I}^e$. Denote the corresponding intervened joint distribution by $P_{\boldsymbol{V}}^e$ with joint density $p^e$ given by

$$p^e(\boldsymbol{v}) = \prod_{i \in \mathcal{I}^e} \tilde{p}_i\left(v_i \mid \boldsymbol{v}_{\mathrm{pa}(i)}\right) \prod_{j \in [n] \setminus \mathcal{I}^e} p_j\left(v_j \mid \boldsymbol{v}_{\mathrm{pa}(j)}\right). \tag{B.4}$$

Denote by $Q_{\boldsymbol{Z}}^e = (\boldsymbol{P}_{\pi^{-1}} \circ \phi)_*(P_{\boldsymbol{V}}^e)$ the corresponding distribution over $\boldsymbol{Z}$ with joint density given by $q^e$

$$q^e(\boldsymbol{z}) = p^e(\psi \circ \boldsymbol{P}_\pi(\boldsymbol{z})) \left| \det \boldsymbol{J}_{\psi \circ \boldsymbol{P}_\pi}(\boldsymbol{z}) \right| \tag{B.5}$$

$$= \prod_{i \in \mathcal{I}^e} \tilde{p}_i\left(\psi_i\left(z_{\pi(i)}\right) \mid \psi_{\mathrm{pa}(i)}\left(\boldsymbol{z}_{\mathrm{pa}(\pi(i); G')}\right)\right) \left| \frac{\mathrm{d}\psi_i}{\mathrm{d}z_{\pi(i)}}\left(z_{\pi(i)}\right) \right| \prod_{j \in [n] \setminus \mathcal{I}^e} q_{\pi(j)}\left(z_{\pi(j)} \mid \boldsymbol{z}_{\mathrm{pa}(\pi(j); G')}\right), \tag{B.6}$$

where we have used (B.3), (B.4), and the fact that $\boldsymbol{J}_\psi$ is diagonal.

By defining

$$\tilde{q}_{\pi(i)}\left(z_{\pi(i)} \mid \boldsymbol{z}_{\mathrm{pa}(\pi(i); G')}\right) := \tilde{p}_i\left(\psi_i\left(z_{\pi(i)}\right) \mid \psi_{\mathrm{pa}(i)}\left(\boldsymbol{z}_{\mathrm{pa}(\pi(i); G')}\right)\right) \left| \frac{\mathrm{d}\psi_i}{\mathrm{d}z_{\pi(i)}}\left(z_{\pi(i)}\right) \right| \tag{B.7}$$

we finally arrive at

$$q^e(\boldsymbol{z}) = \prod_{i \in \mathcal{I}^e} \tilde{q}_{\pi(i)} \left( z_{\pi(i)} \mid \boldsymbol{z}_{\mathrm{pa}(\pi(i);G')} \right) \prod_{j \in [n] \setminus \mathcal{I}^e} q_{\pi(j)} \left( z_{\pi(j)} \mid \boldsymbol{z}_{\mathrm{pa}(\pi(j);G')} \right) . \tag{B.8}$$

This shows that any intervention on $\{V_i\}_{i \in \mathcal{I}^e} \subseteq \boldsymbol{V}$ which replaces

$$\left\{ p_i(v_i \mid \boldsymbol{v}_{\mathrm{pa}(i)}) \right\}_{i \in \mathcal{I}^e} \mapsto \left\{ \tilde{p}_i(v_i \mid \boldsymbol{v}_{\mathrm{pa}(i)}) \right\}_{i \in \mathcal{I}^e} , \tag{B.9}$$

can equivalently be captured by an intervention on $\{Z_{\pi(i)}\}_{i \in \mathcal{I}^e} \subseteq \boldsymbol{Z}$ which replaces

$$\left\{ q_{\pi(i)} \left( z_{\pi(i)} \mid \boldsymbol{z}_{\mathrm{pa}(\pi(i);G')} \right) \right\}_{i \in \mathcal{I}^e} \mapsto \left\{ \tilde{q}_{\pi(i)} \left( z_{\pi(i)} \mid \boldsymbol{z}_{\mathrm{pa}(\pi(i);G')} \right) \right\}_{i \in \mathcal{I}^e} . \tag{B.10}$$

with $\tilde{q}_i$ defined according to (B.7). $\qquad \square$

# C  Identifiability Proofs

## C.1  Auxiliary Lemmata

**Lemma C.1** (Lemma 2 of Brehmer et al. [18]). *Let $A = C = \mathbb{R}$ and $B = \mathbb{R}^n$. Let $f : A \times B \to C$ be differentiable. Define two differentiable measures $p_A$ on $A$ and $p_C$ on $C$. Let $\forall b \in B$, $f(\cdot, b) : A \to C$ be measure-preserving, in the sense that the pushforward of $p_A$ is always $p_C$. Then $f(\cdot, b)$ is constant in $b$ on $B$.*

*Proof.* See Appendix A.2 of Brehmer et al. [18]. $\qquad \square$

**Lemma C.2** (Preservation of conditional independence under invertible transformation.). *Let $X$ and $Y$ be continuous real-valued random variables, and let $\boldsymbol{Z}$ be a continuous random vector taking values in $\mathbb{R}^n$. Suppose that $(X, Y, \boldsymbol{Z})$ have a joint density w.r.t. the Lebesgue measure. Let $f : \mathbb{R} \to \mathbb{R}$, $g : \mathbb{R} \to \mathbb{R}$, and $h : \mathbb{R}^n \to \mathbb{R}^n$ be diffeomorphisms. Then $X \perp\!\!\!\perp Y \mid \boldsymbol{Z} \implies f(X) \perp\!\!\!\perp g(Y) \mid h(\boldsymbol{Z})$.*

*Proof.* Denote by $p(x, y, \boldsymbol{z})$ the density of $(X, Y, \boldsymbol{Z})$. Then $X \perp\!\!\!\perp Y \mid \boldsymbol{Z}$ implies that for all $(x, y, \boldsymbol{z})$, $p$ can be factorized as follows:

$$p(x, y, \boldsymbol{z}) = p_{\boldsymbol{z}}(\boldsymbol{z}) p_x(x \mid \boldsymbol{z}) p_y(y \mid \boldsymbol{z}) . \tag{C.1}$$

Let $(A, B, \boldsymbol{C}) = (f(X), g(Y), h(\boldsymbol{Z}))$, and write $\tilde{f} = f^{-1}$, $\tilde{g} = g^{-1}$, and $\tilde{h} = h^{-1}$.

Then $(A, B, \boldsymbol{C})$ also has a density $q(a, b, \boldsymbol{c})$, which for all $(a, b, \boldsymbol{c})$ is given by the change of variable formula:

$$q(a, b, \boldsymbol{c}) = p\left( \tilde{f}(a), \tilde{g}(b), \tilde{h}(\boldsymbol{c}) \right) \left| \frac{\mathrm{d}\tilde{f}}{\mathrm{d}a}(a) \frac{\mathrm{d}\tilde{g}}{\mathrm{d}b}(b) \det \boldsymbol{J}_{\tilde{h}}(\boldsymbol{c}) \right| \tag{C.2}$$

$$= p_{\boldsymbol{z}} \left( \tilde{h}(\boldsymbol{c}) \right) \left| \det \boldsymbol{J}_{\tilde{h}}(\boldsymbol{c}) \right| p_x \left( \tilde{f}(a) \mid \tilde{h}(\boldsymbol{c}) \right) \left| \frac{\mathrm{d}\tilde{f}}{\mathrm{d}a}(a) \right| p_y \left( \tilde{g}(b) \mid \tilde{h}(\boldsymbol{c}) \right) \left| \frac{\mathrm{d}\tilde{g}}{\mathrm{d}b}(b) \right| \tag{C.3}$$

where in (C.2) we have used the fact that $(X, Y, \boldsymbol{Z}) \mapsto (f(X), g(Y), h(\boldsymbol{Z}))$ has block-diagonal Jacobian, and in (C.3) that $p$ factorises as in (C.1). Next, define

$$q_{\boldsymbol{c}}(\boldsymbol{c}) := p_{\boldsymbol{z}} \left( \tilde{h}(\boldsymbol{c}) \right) \left| \det \boldsymbol{J}_{\tilde{h}}(\boldsymbol{c}) \right| , \tag{C.4}$$

$$q_a(a \mid \boldsymbol{c}) := p_x \left( \tilde{f}(a) \mid \tilde{h}(\boldsymbol{c}) \right) \left| \frac{\mathrm{d}\tilde{f}}{\mathrm{d}a}(a) \right| , \tag{C.5}$$

$$q_b(b \mid \boldsymbol{c}) := p_y \left( \tilde{g}(b) \mid \tilde{h}(\boldsymbol{c}) \right) \left| \frac{\mathrm{d}\tilde{g}}{\mathrm{d}b}(b) \right| . \tag{C.6}$$

Since $p_{\boldsymbol{z}}$, $p_x$, and $p_y$ are valid densities (non-negative and integrating to one), so are $q_{\boldsymbol{c}}$, $q_a$, and $q_b$. Substitution into (C.3) yields for all $(a, b, \boldsymbol{c})$,

$$q(a, b, \boldsymbol{c}) = q_{\boldsymbol{c}}(\boldsymbol{c}) q_a(a \mid \boldsymbol{c}) q_b(b \mid \boldsymbol{c}) , \tag{C.7}$$

which implies that $A \perp\!\!\!\perp B \mid \boldsymbol{C}$, concluding the proof. $\qquad \square$

## C.2 Proof of Thm. 3.2

**Theorem 3.2** (Bivariate identifiability up to $\sim_{\mathrm{CRL}}$ from one perfect stochastic intervention per node). *Suppose that we have access to multiple environments $\{P_{\boldsymbol{X}}^e\}_{e \in \mathcal{E}}$ generated as described in § 2 under Asms. 2.2, 2.5, 2.8 and 2.9 with $n = 2$. Let $(h, G')$ be any candidate solution such that the inferred latent distributions $Q_{\boldsymbol{Z}}^e = h_*(P_{\boldsymbol{X}}^e)$ of $\boldsymbol{Z} = h(\boldsymbol{X})$ and the inferred mixing function $h^{-1}$ satisfy the above assumptions w.r.t. the candidate causal graph $G'$. Assume additionally that*

*(A1) all densities $p^e$ and $q^e$ are continuously differentiable and fully supported on $\mathbb{R}^n$;*

*(A2) we have access to a known observational environment $e_0$ and one single node perfect intervention for each node, with unknown targets: there exist $n + 1$ environments $\mathcal{E} = \{e_i\}_{i=0}^n$ such that $\mathcal{I}^{e_0} = \varnothing$ and for each $i \in [n]$ we have $\mathcal{I}^{e_i} = \{\pi(i)\}$ for an unknown permutation $\pi$ of $[n]$;*

*(A3) for all $i \in [n]$, the intervened mechanisms $\tilde{p}_i(v_i)$ differ from the corresponding base mechanisms $p_i(v_i \mid \boldsymbol{v}_{\mathrm{pa}(i)})$ everywhere, in the sense that*

$$\forall \boldsymbol{v} : \qquad \frac{\partial}{\partial v_i} \frac{\tilde{p}_i(v_i)}{p_i(v_i \mid \boldsymbol{v}_{\mathrm{pa}(i)})} \neq 0 ; \tag{3.1}$$

*(A4) (**"genericity"**) the base and intervened mechanisms are not fine-tuned to each other, in the sense that there exists a continuous function $\varphi : \mathbb{R}^+ \to \mathbb{R}$ for which*

$$\mathbb{E}_{\boldsymbol{v} \sim P_{\boldsymbol{V}}^{e_0}} \left[ \varphi\left( \frac{\tilde{p}_2(v_2)}{p_2(v_2 \mid v_1)} \right) \right] \neq \mathbb{E}_{\boldsymbol{v} \sim P_{\boldsymbol{V}}^{e_1}} \left[ \varphi\left( \frac{\tilde{p}_2(v_2)}{p_2(v_2 \mid v_1)} \right) \right] \tag{3.2}$$

*Then the ground truth is identified in the sense of Defn. 2.6, that is, $(f^{-1}, G) \sim_{\mathrm{CRL}} (h, G')$.*

*Proof.* From the assumption of a shared mixing $f$ and shared encoder $h$ across all environments, we have that

$$\boldsymbol{Z} = h(\boldsymbol{X}) = h(f(\boldsymbol{V})) = h \circ f(\boldsymbol{V}) . \tag{C.8}$$

Let $\psi = f^{-1} \circ h^{-1} : \mathbb{R}^n \to \mathbb{R}^n$ such that

$$\boldsymbol{V} = \psi(\boldsymbol{Z}) .$$

By Asm. 2.5, $f$, $h$, and thus also $h \circ f$ are diffeomorphisms. Hence, $\psi$ is well-defined and also diffeomorphic.

By the change of variable formula, for all $e$ and all $\boldsymbol{z}$ the density $q^e(\boldsymbol{z})$ is given by

$$q^e(\boldsymbol{z}) = p^e(\psi(\boldsymbol{z})) \left| \det \boldsymbol{J}_\psi(\boldsymbol{z}) \right| \tag{C.9}$$

where $(\boldsymbol{J}_\psi(\boldsymbol{z}))_{ij} = \frac{\partial \psi_i}{\partial z_j}(\boldsymbol{z})$ denotes the Jacobian of $\psi$.

We now consider two separate cases, depending on whether the intervention targets in $q^{e_i}$ for $e_i \in \{e_1, e_2\}$ match those in $p^{e_i}$ (Case 1) or not (Case 2).

**Case 1: Aligned Intervention Targets.** According to Asm. 2.8 and (A2), (C.9) applied to the known observational environment $e_0$ and the interventional environments $e_1, e_2$ leads to the system of equations:

$$q_1(z_1) q_2(z_2 \mid z_{\mathrm{pa}(2;G')}) = p_1\left(\psi_1(\boldsymbol{z})\right) p_2\left(\psi_2(\boldsymbol{z}) \mid \psi_{\mathrm{pa}(2)}(\boldsymbol{z})\right) \left| \det \boldsymbol{J}_\psi(\boldsymbol{z}) \right| \tag{C.10}$$

$$\tilde{q}_1(z_1) q_2(z_2 \mid z_{\mathrm{pa}(2;G')}) = \tilde{p}_1\left(\psi_1(\boldsymbol{z})\right) p_2\left(\psi_2(\boldsymbol{z}) \mid \psi_{\mathrm{pa}(2)}(\boldsymbol{z})\right) \left| \det \boldsymbol{J}_\psi(\boldsymbol{z}) \right| \tag{C.11}$$

$$q_1(z_1) \tilde{q}_2(z_2) = p_1\left(\psi_1(\boldsymbol{z})\right) \tilde{p}_2\left(\psi_2(\boldsymbol{z})\right) \left| \det \boldsymbol{J}_\psi(\boldsymbol{z}) \right| \tag{C.12}$$

where $z_{\mathrm{pa}(2;G')}$ denotes the parents of $z_2$ in $G'$, and $\mathrm{pa}(2)$ denotes the parents of $V_2$ in $G$.

Note that neither side of the previous equations can be zero due to the full support assumption[12] (A1) and $\psi$ being diffeomorphic implying the determinant is non-zero.

---

[12]This can also be relaxed to fully supported on a Cartesian product of intervals.

Taking quotients of (C.11) and (C.10), yields

$$\frac{\tilde{q}_1}{q_1}(z_1) = \frac{\tilde{p}_1}{p_1}(\psi_1(\boldsymbol{z})) \,. \tag{C.13}$$

Next, we take the partial derivative w.r.t. $z_2$ on both sides and use the chain rule to obtain:

$$0 = \left(\frac{\tilde{p}_1}{p_1}\right)' (\psi_1(\boldsymbol{z})) \, \frac{\partial \psi_1}{\partial z_2}(\boldsymbol{z}) \,. \tag{C.14}$$

Now, by assumption (A3), the first term on the RHS of (C.14) is non-zero everywhere. Hence,

$$\forall \boldsymbol{z}: \qquad \frac{\partial \psi_1}{\partial z_2}(\boldsymbol{z}) = 0 \,. \tag{C.15}$$

It follows that $\psi_1$ is, in fact, a scalar function, and

$$V_1 = \psi_1(Z_1) \,. \tag{C.16}$$

Since $\psi$ is a diffeomorphism, $\psi_1$ must also be diffeomorphic. Hence, by the change of variable formula applied to (C.16), the marginal density $q_1(z_1)$ is given by

$$q_1(z_1) = p_1(\psi_1(z_1)) \left| \frac{\partial \psi_1}{\partial z_1}(z_1) \right| \,. \tag{C.17}$$

Further, since $\boldsymbol{J}_\psi$ is triangular due to (C.15), its determinant is given by

$$\left| \det \boldsymbol{J}_\psi(\boldsymbol{z}) \right| = \left| \frac{\partial \psi_1}{\partial z_1}(z_1) \, \frac{\partial \psi_2}{\partial z_2}(z_1, z_2) \right| \,. \tag{C.18}$$

Substituting (C.17) and (C.18) into (C.12) yields (after cancellation of equal terms):

$$\tilde{q}_2(z_2) = \tilde{p}_2 \left( \psi_2(z_1, z_2) \right) \left| \frac{\partial \psi_2}{\partial z_2}(z_1, z_2) \right| \,. \tag{C.19}$$

The expression in (C.19) implies that, for all $z_1$, the mapping $\psi_2(z_1, \cdot) : \mathbb{R} \to \mathbb{R}$ is measure preserving for the differentiable $\tilde{q}_2$ and $\tilde{p}_2$. By Lemma C.1 (Lemma 2 of Brehmer et al. [18, § A.2]), it then follows that $\psi_2$ must, in fact, be constant in $z_1$, that is

$$\forall \boldsymbol{z}: \qquad \frac{\partial \psi_2}{\partial z_1}(\boldsymbol{z}) = 0 \,. \tag{C.20}$$

Note that this last step is where the assumption of perfect interventions (Asm. 2.9) is leveraged: the conclusion would not hold for arbitrary imperfect interventions for which (3.8) would involve $\tilde{q}_2(z_2 \mid z_1)$ and $p_2 \left( \psi_2(z_1, z_2) \mid \psi_1(z_1) \right)$.

Hence, we have shown that $\psi$ is an element-wise function:

$$\boldsymbol{V} = (V_1, V_2) = \psi(\boldsymbol{Z}) = (\psi_1(Z_1), \psi_2(Z_2)) \,. \tag{C.21}$$

Finally, since $\psi$ is a diffeomorphism, (C.21) implies that

$$V_1 \perp\!\!\!\perp V_2 \iff Z_1 \perp\!\!\!\perp Z_2 \,. \tag{C.22}$$

It then follows from the faithfulness assumption (Asm. 2.2) that we also must have $G = G'$.

This concludes the proof of Case 1, as we have shown that $(h, G') \sim_{\mathrm{CRL}} (f^{-1}, G)$ with $G' = \pi(G) = G$ ($\pi$ being the identity permutation) and $h \circ f = \psi^{-1} =: \phi$ an element-wise diffeomorphism.

**Case 2: Misaligned Intervention Targets.** Writing down the system of equations similar to (C.10)–(C.12), but for the case with misaligned intervention targets across $p$ and $q$ yields:

$$q_1(z_1) q_2(z_2 \mid z_{\mathrm{pa}(2;G')}) = p_1 \left( \psi_1(\boldsymbol{z}) \right) p_2 \left( \psi_2(\boldsymbol{z}) \mid \psi_{\mathrm{pa}(2)}(\boldsymbol{z}) \right) \left| \det \boldsymbol{J}_\psi(\boldsymbol{z}) \right| \tag{C.23}$$

$$\tilde{q}_1(z_1) q_2(z_2 \mid z_{\mathrm{pa}(2;G')}) = p_1 \left( \psi_1(\boldsymbol{z}) \right) \tilde{p}_2 \left( \psi_2(\boldsymbol{z}) \right) \left| \det \boldsymbol{J}_\psi(\boldsymbol{z}) \right| \tag{C.24}$$

$$q_1(z_1) \tilde{q}_2(z_2) = \tilde{p}_1 \left( \psi_1(\boldsymbol{z}) \right) p_2 \left( \psi_2(\boldsymbol{z}) \mid \psi_{\mathrm{pa}(2)}(\boldsymbol{z}) \right) \left| \det \boldsymbol{J}_\psi(\boldsymbol{z}) \right| \,. \tag{C.25}$$

Taking ratios of $e_1$ and $e_2$ with $e_0$ yields

$$\frac{\tilde{q}_1}{q_1}(z_1) = \frac{\tilde{p}_2\left(\psi_2(\boldsymbol{z})\right)}{p_2\left(\psi_2(\boldsymbol{z}) \mid \psi_{\mathrm{pa}(2)}(\boldsymbol{z})\right)} \tag{C.26}$$

$$\frac{\tilde{q}_2(z_2)}{q_2(z_2 \mid z_{\mathrm{pa}(2;G')})} = \frac{\tilde{p}_1}{p_1}\left(\psi_1(\boldsymbol{z})\right) . \tag{C.27}$$

We separate the remainder of the proof of Case 2 into different subcases depending on $G$ and $G'$: as we will see, we can use a similar reasoning as in Case 1, except for the case where both $G$ and $G'$ are missing no edge.

*Case 2a: $V_1 \not\rightarrow V_2$ in $G$, that is $\mathrm{pa}(2) = \varnothing$.* Then we can proceed similarly to Case 1. First, we take the partial derivative of (C.26) w.r.t. $z_2$ to arrive at:

$$0 = \left(\frac{\tilde{p}_2}{p_2}\right)'\left(\psi_2(\boldsymbol{z})\right)\frac{\partial \psi_2}{\partial z_2}(\boldsymbol{z}) . \tag{C.28}$$

Using (A3), this implies that $\psi_2$ does not depend on $Z_2$, that is, $V_2 = \psi_2(Z_1)$.

Next, we again write $q(z_1)$ in terms of $p_2(\psi_2(z_1))$ using the univariate change of variable formula, substitute into (C.25), cancel the corresponding terms, and arrive at:

$$\tilde{q}_2(z_2) = \tilde{p}_1\left(\psi_1(z_1, z_2)\right)\left|\frac{\partial \psi_1}{\partial z_2}(z_1, z_2)\right| \tag{C.29}$$

Lemma C.1 applied to $\psi_1(z_1, \cdot)$ which preserves $\tilde{q}_2$ and $\tilde{p}_1$ for all $z_1$ shows that $\psi_1$ is constant in $Z_1$, that is

$$\boldsymbol{V} = (V_1, V_2) = \psi(\boldsymbol{Z}) = (\psi_1(Z_2), \psi_2(Z_1)) . \tag{C.30}$$

Since $V_1 \perp\!\!\!\perp V_2$ by the assumption of Case 2a, it follows from the invertible element-wise reparametrisation above that $Z_1 \perp\!\!\!\perp Z_2$ and hence, by faithfulness, $Z_1 \not\rightarrow Z_2$ in $G'$.

Finally, note that there is no partial order on the empty graph and so $G' = \pi(G) = G$ and $\boldsymbol{Z} = \boldsymbol{P}_{\pi^{-1}} \cdot \psi^{-1}(\boldsymbol{V})$ where $\pi$ is the nontrivial permutation of $\{1, 2\}$.

*Case 2b: $V_1 \rightarrow V_2$ in $G$, that is $\mathrm{pa}(2) = \{1\}$.* If $G' \neq G$, that is $Z_1 \not\rightarrow Z_2$ in $G'$, then the same argument as in Case 2a, this time starting by taking the partial derivative of (C.27) w.r.t. $z_1$, can be used to reach the same conclusion in (C.30). However, this contradicts faithfulness since $V_1 \not\!\perp\!\!\!\perp V_2$ in $G$.

Hence, we must have $G' = G$, and the following two equations must hold for all $\boldsymbol{z}$:

$$\frac{\tilde{q}_1(z_1)}{q_1(z_1)} = \frac{\tilde{p}_2\left(\psi_2(\boldsymbol{z})\right)}{p_2\left(\psi_2(\boldsymbol{z}) \mid \psi_1(\boldsymbol{z})\right)} \tag{C.31}$$

$$\frac{\tilde{q}_2(z_2)}{q_2(z_2 \mid z_1)} = \frac{\tilde{p}_1\left(\psi_1(\boldsymbol{z})\right)}{p_1\left(\psi_1(\boldsymbol{z})\right)} \tag{C.32}$$

The remainder of the proof consists of exploring the implications of (C.32) and (C.31), ultimately resulting in a violation of the genericity condition (A4).

To ease notation, define the following auxiliary functions:

$$a(z_1) := \frac{\tilde{q}_1(z_1)}{q_1(z_1)} , \tag{C.33}$$

$$b(\boldsymbol{v}) := \frac{\tilde{p}_2(v_2)}{p_2(v_2 \mid v_1)} , \tag{C.34}$$

$$c(\boldsymbol{z}) := \frac{\tilde{q}_2(z_2)}{q_{2|1}(z_2 \mid z_1)} , \tag{C.35}$$

$$d(v_1) := \frac{\tilde{p}_1(v_1)}{p_1(v_1)} . \tag{C.36}$$

With this, (C.31) and (C.32) take the following form:

$$a(z_1) = b(\psi(\boldsymbol{z})).$$ (C.37)

$$c(\boldsymbol{z}) = d(\psi_1(\boldsymbol{z})),$$ (C.38)

Next, define the following maps:

$$\kappa : \boldsymbol{z} \mapsto \begin{bmatrix} a(z_1) \\ c(\boldsymbol{z}) \end{bmatrix}$$ (C.39)

$$\rho : \boldsymbol{v} \mapsto \begin{bmatrix} b(\boldsymbol{v}) \\ d(v_1) \end{bmatrix}$$ (C.40)

Then, (C.37) and (C.38) together imply that

$$\kappa = \rho \circ \psi.$$ (C.41)

Recalling that by (A1) all densities are continuously differentiable, the Jacobians of $\kappa$ and $\rho$ are given by:

$$\boldsymbol{J}_\kappa(\boldsymbol{z}) = \begin{bmatrix} a'(z_1) & 0 \\ \frac{\partial c}{\partial z_1}(\boldsymbol{z}) & \frac{\partial c}{\partial z_2}(\boldsymbol{z}) \end{bmatrix},$$ (C.42)

$$\boldsymbol{J}_\rho(\boldsymbol{v}) = \begin{bmatrix} \frac{\partial b}{\partial v_1}(\boldsymbol{v}) & \frac{\partial b}{\partial v_2}(\boldsymbol{v}) \\ d'(v_1) & 0 \end{bmatrix},$$ (C.43)

and the corresponding determinants are given by

$$\left| \det \boldsymbol{J}_\kappa(\boldsymbol{z}) \right| = \left| a'(z_1) \frac{\partial c}{\partial z_2}(\boldsymbol{z}) \right| \neq 0$$ (C.44)

$$\left| \det \boldsymbol{J}_\rho(\boldsymbol{v}) \right| = \left| d'(v_1) \frac{\partial b}{\partial v_2}(\boldsymbol{v}) \right| \neq 0$$ (C.45)

where the inequalities for all $\boldsymbol{z}$ follow since, by assumption (A3), the derivatives of ratios of intervened and original mechanisms are non-vanishing everywhere:

$$a'(z_1) \neq 0 \neq \frac{\partial c}{\partial z_2}(\boldsymbol{z}) \qquad \text{and} \qquad d'(v_1) \neq 0 \neq \frac{\partial b}{\partial v_2}(\boldsymbol{v}),$$ (C.46)

This implies that the following families of maps are continuously differentiable, monotonic, and invertible,

$$a : z_1 \mapsto a(z_1),$$ (C.47)

$$b_{v_1} : v_2 \mapsto b(v_1, v_2),$$ (C.48)

$$c_{z_1} : z_2 \mapsto c(z_1, z_2),$$ (C.49)

$$d : v_1 \mapsto d(v_1),$$ (C.50)

with continuously differentiable inverses

$$a^{-1} : w_1 \mapsto a^{-1}(w_1),$$ (C.51)

$$b_{v_1}^{-1} : w_1 \mapsto b_{v_1}^{-1}(w_1),$$ (C.52)

$$c_{z_1}^{-1} : w_2 \mapsto c_{z_1}^{-1}(w_2),$$ (C.53)

$$d^{-1} : w_2 \mapsto d^{-1}(w_2).$$ (C.54)

This implies that $\rho$ and $\kappa$ are valid diffeomorphisms onto their image and their inverses are given by:

$$\kappa^{-1} : \boldsymbol{w} \mapsto \begin{bmatrix} a^{-1}(w_1) \\ c_{a^{-1}(w_1)}^{-1}(w_2) \end{bmatrix},$$ (C.55)

$$\rho^{-1} : \boldsymbol{w} \mapsto \begin{bmatrix} d^{-1}(w_2) \\ b_{d^{-1}(w_2)}^{-1}(w_1) \end{bmatrix}.$$ (C.56)

Since $\boldsymbol{V} = \psi(\boldsymbol{Z})$, by (C.41) we have

$$\boldsymbol{W} := \rho(\boldsymbol{V}) = \rho \circ \psi(\boldsymbol{Z}) = \kappa(\boldsymbol{Z}). \tag{C.57}$$

Denote the distributions of $\boldsymbol{W}$ by $R_{\boldsymbol{W}}$ and its density by $r(\boldsymbol{w})$. Since for all $e$, we have

$$P_{\boldsymbol{V}}^e = \psi_*(Q_{\boldsymbol{Z}}^e) \tag{C.58}$$

it follows from (C.57) that

$$R_{\boldsymbol{W}}^e = \rho_*(P_{\boldsymbol{V}}^e) = \kappa_*(Q_{\boldsymbol{Z}}^e). \tag{C.59}$$

This provides two different ways of applying the change of variable formula to compute $r(\boldsymbol{w})$.

First, we consider the pushforward of $Q_{\boldsymbol{Z}}^{e_0}$ by $\kappa$:

$$r(\boldsymbol{w}) = q\left(\kappa^{-1}(\boldsymbol{w})\right) \left|\det \boldsymbol{J}_{\kappa^{-1}}(\boldsymbol{w})\right| \tag{C.60}$$

$$= q_1\left(a^{-1}(w_1)\right) q_2\left(c_{a^{-1}(w_1)}^{-1}(w_2) \mid a^{-1}(w_1)\right) \left|\frac{\mathrm{d}}{\mathrm{d}w_1}a^{-1}(w_1)\frac{\mathrm{d}}{\mathrm{d}w_2}c_{a^{-1}(w_1)}^{-1}(w_2)\right| \tag{C.61}$$

By integrating this joint density with respect to $w_2$, we obtain the following expression for the marginal $r_1(w_1)$:

$$r_1(w_1) = \left|\frac{\mathrm{d}}{\mathrm{d}w_1}a^{-1}(w_1)\right| q_1\left(a^{-1}(w_1)\right) \int q_2\left(c_{a^{-1}(w_1)}^{-1}(w_2) \mid a^{-1}(w_1)\right) \left|\frac{\mathrm{d}}{\mathrm{d}w_2}c_{a^{-1}(w_1)}^{-1}(w_2)\right| \mathrm{d}w_2. \tag{C.62}$$

By the diffeomorphic change of variable $z_2 = c_{a^{-1}(w_1)}^{-1}(w_2)$, [13] this can be written as

$$r_1(w_1) = \left|\frac{d}{dw_1}a^{-1}(w_1)\right| q_1\left(a^{-1}(w_1)\right) \int q_2\left(z_2 \mid a^{-1}(w_1)\right) dz_2 \tag{C.63}$$

$$= \left|\frac{d}{dw_1}a^{-1}(w_1)\right| q_1\left(a^{-1}(w_1)\right) \tag{C.64}$$

Next, we carry out the same calculation for the pushforward of $P_{\boldsymbol{V}}^{e_0}$ by $\rho$:

$$r(\boldsymbol{w}) = p\left(\rho^{-1}(\boldsymbol{w})\right) \left|\det \boldsymbol{J}_{\rho^{-1}}(\boldsymbol{w})\right| \tag{C.65}$$

$$= p_1\left(d^{-1}(w_2)\right) p_2\left(b_{d^{-1}(w_2)}^{-1}(w_1) \mid d^{-1}(w_2)\right) \left|\frac{\mathrm{d}}{\mathrm{d}w_2}d^{-1}(w_2)\frac{\mathrm{d}}{\mathrm{d}w_1}b_{d^{-1}(w_2)}^{-1}(w_1)\right|, \tag{C.66}$$

leading to the marginal

$$r_1(w_1) = \int p_1\left(d^{-1}(w_2)\right) p_2\left(b_{d^{-1}(w_2)}^{-1}(w_1) \mid d^{-1}(w_2)\right) \left|\frac{\mathrm{d}}{\mathrm{d}w_1}b_{d^{-1}(w_2)}^{-1}(w_1)\right| \left|\frac{\mathrm{d}}{\mathrm{d}w_2}d^{-1}(w_2)\right| \mathrm{d}w_2 \tag{C.67}$$

$$= \int p_1(v_1)p_2\left(b_{v_1}^{-1}(w_1) \mid v_1\right) \left|\frac{\mathrm{d}}{\mathrm{d}w_1}b_{v_1}^{-1}(w_1)\right| \mathrm{d}v_1, \tag{C.68}$$

where the second line is obtained by the diffeomorphic change of variable $v_1 = d^{-1}(w_2)$.

Equating the two expressions for $r(w_1)$ in $e_0$ in (C.68) and (C.64), we obtain for all $w_1$:

$$\left|\frac{\mathrm{d}}{\mathrm{d}w_1}a^{-1}(w_1)\right| q_1\left(a^{-1}(w_1)\right) = \int p_1(v_1)p_2\left(b_{v_1}^{-1}(w_1) \mid v_1\right) \left|\frac{\mathrm{d}}{\mathrm{d}w_1}b_{v_1}^{-1}(w_1)\right| \mathrm{d}v_1. \tag{C.69}$$

Applying the same approach to the environment in which $V_1$ is intervened upon changing $p_1$ to $\tilde{p}_1$ while $Z_2$ is intervened upon leaving $q_1$ invariant, yields for all $w_1$:

$$\left|\frac{\mathrm{d}}{\mathrm{d}w_1}a^{-1}(w_1)\right| q_1\left(a^{-1}(w_1)\right) = \int \tilde{p}_1(v_1)p_2\left(b_{v_1}^{-1}(w_1) \mid v_1\right) \left|\frac{\mathrm{d}}{\mathrm{d}w_1}b_{v_1}^{-1}(w_1)\right| \mathrm{d}v_1. \tag{C.70}$$

---

[13] Note that: $\int q_2(z_2(w_2)) \left|\frac{\mathrm{d}z_2}{\mathrm{d}w_2}\right| \mathrm{d}w_2 = \int q_2(z_2)\, \mathrm{d}z_2$.

Finally, by equating (C.69) and (C.70), we arrive at the following expression which must hold for all $w_1$:

$$\int p_1(v_1) p_2\left(b_{v_1}^{-1}(w_1) \mid v_1\right) \left|\frac{\mathrm{d}}{\mathrm{d}w_1} b_{v_1}^{-1}(w_1)\right| \mathrm{d}v_1 = \int \tilde{p}_1(v_1) p_2\left(b_{v_1}^{-1}(w_1) \mid v_1\right) \left|\frac{\mathrm{d}}{\mathrm{d}w_1} b_{v_1}^{-1}(w_1)\right| \mathrm{d}v_1 \tag{C.71}$$

which we can rewrite as

$$\int \left(\tilde{p}_1(v_1) - p_1(v_1)\right) p_2\left(b_{v_1}^{-1}(w_1) \mid v_1\right) \left|\frac{\mathrm{d}}{\mathrm{d}w_1} b_{v_1}^{-1}(w_1)\right| \mathrm{d}v_1 = 0 \,. \tag{C.72}$$

Multiplying by any continuous function $\varphi(w_1)$, integrating w.r.t. $w_1$ and applying the diffeomorphic change of variable $v_2 = b_{v_1}^{-1}(w_1)$, this can be expressed as:

$$0 = \int \varphi(w_1) \int \left(\tilde{p}_1(v_1) - p_1(v_1)\right) p_2\left(b_{v_1}^{-1}(w_1) \mid v_1\right) \left|\frac{\mathrm{d}}{\mathrm{d}w_1} b_{v_1}^{-1}(w_1)\right| \mathrm{d}v_1 \, \mathrm{d}w_1 \tag{C.73}$$

$$= \int\int \varphi\left(b_{v_1}(v_2)\right) \left(\tilde{p}_1(v_1) - p_1(v_1)\right) p_2(v_2 \mid v_1) \, \mathrm{d}v_2 \, \mathrm{d}v_1 \tag{C.74}$$

$$= \int\int \varphi\left(\frac{\tilde{p}_2(v_2)}{p_2(v_2 \mid v_1)}\right) \left(\tilde{p}_1(v_1) - p_1(v_1)\right) p_2(v_2 \mid v_1) \, \mathrm{d}v_2 \, \mathrm{d}v_1 \tag{C.75}$$

where we have resubstituted the expression for $b_{v_1}(v_2)$ in the last line.

Equivalently, this can be written as: *for any continuous function $\varphi$,*

$$\mathbb{E}_{\boldsymbol{v}\sim P_{\boldsymbol{V}}^{e_0}}\left[\varphi\left(\frac{\tilde{p}_2(v_2)}{p_2(v_2 \mid v_1)}\right)\right] = \mathbb{E}_{\boldsymbol{v}\sim P_{\boldsymbol{V}}^{e_1}}\left[\varphi\left(\frac{\tilde{p}_2(v_2)}{p_2(v_2 \mid v_1)}\right)\right] \,. \tag{C.76}$$

However, the genericity condition (A4) precisely rules this out, since the above equality must be violated for at least one $\varphi$, concluding this last case.

To sum up, all cases either lead to a contradiction, or imply the conclusion that $(f^{-1}, G) \sim_{\mathrm{CRL}} (h, G')$, concluding the proof. $\qquad\square$

## C.3 Proof of Thm. 3.4

**Theorem 3.4** (Identifiability up to $\sim_{\mathrm{CRL}}$ from two paired perfect stochastic interventions per node).
*Suppose that we have access to multiple environments $\{P_{\boldsymbol{X}}^e\}_{e\in\mathcal{E}}$ generated as described in § 2 under Asms. 2.2, 2.3, 2.5, 2.8 and 2.9. Let $(h, G')$ be any candidate solution such that the inferred latent distributions $Q_{\boldsymbol{Z}}^e = h_*(P_{\boldsymbol{X}}^e)$ of $\boldsymbol{Z} = h(\boldsymbol{X})$ and the inferred mixing function $h^{-1}$ satisfy the above assumptions w.r.t. the candidate causal graph $G'$. Assume additionally that*

*(A1) all densities $p^e$ and $q^e$ are continuously differentiable and fully supported on $\mathbb{R}^n$;*

*(A2') we have access to at least one* pair *of single-node perfect interventions per node, with unknown targets: there exist $m \geq n$ known pairs of environments $\mathcal{E} = \{(e_j, e_j')\}_{j=1}^m$ such that for each $i \in [n]$ there exists some* unknown *$j \in [m]$ for which $\mathcal{I}^{e_j} = \mathcal{I}^{e_j'} = \{i\}$;*

*(A3') for all $i \in [n]$, the intervened mechanisms $\tilde{p}_i(v_i)$ and $\tilde{\tilde{p}}_i(v_i)$ differ everywhere, in the sense that*

$$\forall v_i : \qquad \left(\frac{\tilde{\tilde{p}}_i}{\tilde{p}_i}\right)'(v_i) \neq 0 \,; \tag{3.10}$$

*Then the ground truth is identified in the sense of Defn. 2.6, that is, $(f^{-1}, G) \sim_{\mathrm{CRL}} (h, G')$.*

*Proof.* First, we show that we can extract from the $m \geq n$ available pairs of environments a suitable subset $\mathcal{E}_n$ of exactly $n$ pairs, containing one pair of interventional environments for each node.

Let $\mathcal{E}_n \subseteq \mathcal{E}$ be a subset of $n$ pairs of environments which are assumed to correspond to distinct targets in the model $q$, and suppose for a contradiction that this is not actually the case for the ground truth $p$ (i.e., there are duplicate and missing interventions w.r.t. $p$). Then there must be two pairs of environments $(e_a, e_a'), (e_b, e_b') \in \mathcal{E}_n$, both corresponding to interventions on some $V_i$ in $p$, but which

are modelled as interventions on distinct nodes $Z_j$ and $Z_k$ with $j \neq k$ in $q$. We show that this implies that $V_i$ must simultaneously be a deterministic function of only $Z_j$ and only $Z_k$. Similar to the proof of Thm. 3.2, we obtain the following equations,

$$\frac{\tilde{\tilde{q}}_j}{\tilde{q}_j}\left(z_j\right) = \frac{\tilde{\tilde{p}}_i}{\tilde{p}_i}\left(\psi_i(\boldsymbol{z})\right), \tag{C.77}$$

$$\frac{\tilde{\tilde{q}}_k}{\tilde{q}_k}\left(z_k\right) = \frac{\hat{\tilde{p}}_i}{\hat{p}_i}\left(\psi_i(\boldsymbol{z})\right). \tag{C.78}$$

By taking partial derivatives w.r.t. $z_l$ and applying assumption (A3'), we find that

$$\frac{\partial \psi_i}{\partial z_l} = 0 \qquad \forall l \neq j, \tag{C.79}$$

$$\frac{\partial \psi_i}{\partial z_l} = 0 \qquad \forall l \neq k. \tag{C.80}$$

Since $j \neq k$, this implies that $\partial \psi_i / \partial z_l = 0$ for all $l$ which contradicts invertibility of $\psi$. Thus, by contradiction, we find that $\mathcal{E}_n$ must contain exactly one pair of intervention per node also w.r.t. $p$. For the remainder of the proof, we only consider $\mathcal{E}_n$.

W.l.o.g., for any $(e_i, e'_i) \in \mathcal{E}_n$ we now fix the intervention targets in $p$ to $\mathcal{I}^{e_i} = \mathcal{I}^{e'_i} = \{i\}$ and let $\pi$ be a permutation of $[n]$ such that $\pi(i)$ denotes the inferred intervention target in $q$ that by (A2') is shared across $(e_i, e'_i)$. (We will show later that not all permutations are admissible, but only ones that preserve the partial order of $G$.)

The first part of the proof is similar to Case 1 in the proof of Thm. 3.2. Consider the densities in environments $e_i$ and $e'_i$, which are related through the change of variable formula by:

$$\tilde{q}_{\pi(i)}\left(z_{\pi(i)}\right) \prod_{j \in [n] \setminus \{\pi(i)\}} q_j\left(z_j \mid \boldsymbol{z}_{\mathrm{pa}(j;G')}\right) = \tilde{p}_i\left(\psi_i(\boldsymbol{z})\right) \prod_{j \in [n] \setminus \{i\}} p_j\left(\psi_j(\boldsymbol{z}) \mid \psi_{\mathrm{pa}(j)}(\boldsymbol{z})\right) \left|\det \boldsymbol{J}_\psi(\boldsymbol{z})\right|, \tag{C.81}$$

$$\tilde{\tilde{q}}_{\pi(i)}\left(z_{\pi(i)}\right) \prod_{j \in [n] \setminus \{\pi(i)\}} q_j\left(z_j \mid \boldsymbol{z}_{\mathrm{pa}(j;G')}\right) = \tilde{\tilde{p}}_i\left(\psi_i(\boldsymbol{z})\right) \prod_{j \in [n] \setminus \{i\}} p_j\left(\psi_j(\boldsymbol{z}) \mid \psi_{\mathrm{pa}(j)}(\boldsymbol{z})\right) \left|\det \boldsymbol{J}_\psi(\boldsymbol{z})\right|, \tag{C.82}$$

where $\boldsymbol{Z}_{\mathrm{pa}(j;G')} \subseteq \boldsymbol{Z} \setminus \{Z_j\}$ denotes the parents of $Z_j$ in $G'$.

Taking the quotient of the two equations yields

$$\frac{\tilde{\tilde{q}}_{\pi(i)}}{\tilde{q}_{\pi(i)}}\left(z_{\pi(i)}\right) = \frac{\tilde{\tilde{p}}_i}{\tilde{p}_i}\left(\psi_i(\boldsymbol{z})\right). \tag{C.83}$$

Next, for any $j \neq \pi(i)$, taking partial derivatives w.r.t. $z_j$ on both sides yields

$$0 = \left(\frac{\tilde{\tilde{p}}_i}{\tilde{p}_i}\right)'\left(\psi_i(\boldsymbol{z})\right) \frac{\partial \psi_i}{\partial z_j}(\boldsymbol{z}). \tag{C.84}$$

By assumption (A3'), the first term on the RHS is non-zero everywhere. Hence, (C.84) implies

$$\forall j \neq \pi(i), \forall \boldsymbol{z}: \quad \frac{\partial \psi_i}{\partial z_j}(\boldsymbol{z}) = 0 \tag{C.85}$$

from which we can conclude that

$$V_i = \psi_i\left(Z_{\pi(i)}\right) \tag{C.86}$$

for all $i \in [n]$. That is, $\psi$ is the composition of the permutation $\pi$ with an element-wise reparametrisation.

It remains to show that $\pi$ must, in fact, be a graph isomorphism, which is equivalent to the statement

$$V_i \to V_j \quad \text{in} \quad G \quad \Longleftrightarrow \quad Z_{\pi(i)} \to Z_{\pi(j)} \quad \text{in} \quad G'. \tag{C.87}$$

( $\implies$ ) Suppose for a contradiction that there exist $(i, j)$ such that $V_i \to V_j$ in $G$, but $Z_{\pi(i)} \not\to Z_{\pi(j)}$ in $G'$.

The main idea is to demonstrate that the lack of such direct arrow implies a certain conditional independence which, by faithfulness, would contradict the unconditional dependence of $V_i$ and $V_j$.

Consider environment $e_i$ in which there are perfect interventions on $Z_{\pi(i)}$ and $V_i$, which has the effect of removing all incoming arrows to $Z_{\pi(i)}$ and $V_i$ in the respective post-intervention graphs $G'_{\overline{Z}_{\pi(i)}}$ and $G_{\overline{V}_i}$.

As a result of this and the lack of direct arrow by assumption, any d-connecting path between $Z_{\pi(i)}$ and $Z_{\pi(j)}$ must enter the latter via $\boldsymbol{Z}_{\mathrm{pa}(\pi(j);G')}$ [95].

It then follows from Markovianity of $q$ w.r.t. $G'$ that the following holds in $Q_{\boldsymbol{Z}}^{e_i}$:

$$Z_{\pi(i)} \perp\!\!\!\perp Z_{\pi(j)} \mid \boldsymbol{Z}_{\mathrm{pa}(\pi(j);G')} . \tag{C.88}$$

We now consider the corresponding implication for $P_{\boldsymbol{V}}^{e_i}$. Define

$$\tilde{\boldsymbol{V}} = \left\{ V_k = \psi_k \left( Z_{\pi(k)} \right) : Z_{\pi(k)} \in \boldsymbol{Z}_{\mathrm{pa}(\pi(j);G')} \right\} \subseteq \boldsymbol{V} \setminus \{V_i, V_j\}, \tag{C.89}$$

and note that by assumption, $Z_{\pi(i)} \notin \boldsymbol{Z}_{\mathrm{pa}(\pi(j);G')}$ and hence $V_i \notin \tilde{\boldsymbol{V}}$.

By applying the corresponding diffeomorphic functions $\psi_i$ from (C.86) to (C.88), it follows from Lemma C.2 that

$$V_i \perp\!\!\!\perp V_j \mid \tilde{\boldsymbol{V}} \tag{C.90}$$

in $P_{\boldsymbol{V}}^{e_i}$. However, this violates faithfulness (Asm. 2.2) of $P_{\boldsymbol{V}}$ to $G$ since $V_i$ and $V_j$ are d-connected in $G_{\overline{V}_i}$.

Thus, by contradiction, we must have $Z_{\pi(i)} \to Z_{\pi(j)}$ in $G'$.

( $\impliedby$ ) Now, suppose for a contradiction that there exist $(i, j)$ such that $Z_{\pi(i)} \to Z_{\pi(j)}$ in $G'$, but $V_i \not\to V_j$ in $G$.

By the same argument as before, we find that

$$V_i \perp\!\!\!\perp V_j \mid \boldsymbol{V}_{\mathrm{pa}(j)} \tag{C.91}$$

in $P_{\boldsymbol{V}}^{e_i}$, and thus by Lemma C.2

$$Z_{\pi(i)} \perp\!\!\!\perp Z_{\pi(j)} \mid \tilde{\boldsymbol{Z}} \tag{C.92}$$

in $Q_{\boldsymbol{Z}}^{e_i}$ where

$$\tilde{\boldsymbol{Z}} = \left\{ Z_{\pi(k)} : V_k \in \boldsymbol{V}_{\mathrm{pa}(j)} \right\} \subseteq \boldsymbol{Z} \setminus \{Z_{\pi(i)}, Z_{\pi(j)}\} .$$

However, this contradicts faithfulness of $Q_{\boldsymbol{Z}}$ to $G'$. Hence, we must have that $V_i \to V_j$ in $G$.

This shows that $\pi$ must be a graph isomorphism, thus concluding the proof. $\qquad\square$

### C.4 Proof of Thm. 4.2

**Theorem 4.2** (Preservation of causal influences under $\sim_{\mathrm{CRL}}$)**.** *Let $P_{\boldsymbol{V}}$ be Markovian w.r.t. $G$, let $\pi$ be a graph isomorphism of $G$, and let $\phi$ be an element-wise diffeomorphism. Let $\boldsymbol{Z} = \boldsymbol{P}_{\pi^{-1}} \circ \phi(\boldsymbol{V})$ and denote its induced distribution by $Q_{\boldsymbol{Z}}$. Then for any $V_i \to V_j$ in $G$ we have $\mathfrak{C}_{i \to j}^{P_{\boldsymbol{V}}} = \mathfrak{C}_{\pi(i) \to \pi(j)}^{Q_{\boldsymbol{Z}}}$.*

*Proof.* First, recall that according to Defn. 4.1,

$$\mathfrak{C}_{i \to j}^{P_{\boldsymbol{V}}} := D_{\mathrm{KL}}\left( P_{\boldsymbol{V}} \,\|\, P_{\boldsymbol{V}}^{i \to j} \right), \tag{C.93}$$

where $P_{\boldsymbol{V}}^{i \to j}$ denotes the interventional distribution obtained by replacing $p_j \left( v_j \mid \boldsymbol{v}_{\mathrm{pa}(j)} \right)$ with

$$p_j^{i \to j} \left( v_j \mid \boldsymbol{v}_{\mathrm{pa}(j) \setminus \{i\}} \right) = \int_{\mathcal{V}_i} p_j \left( v_j \mid \boldsymbol{v}_{\mathrm{pa}(j)} \right) p_i(v_i) \, \mathrm{d}v_i . \tag{C.94}$$

Writing out the KL divergence and noting that all terms except the interved mechanism $j$ cancel inside the log, we obtain

$$\mathfrak{C}_{i\to j}^{P\boldsymbol{v}} = \int_{\mathcal{V}} \log\left(\frac{p_j\left(v_j \mid \boldsymbol{v}_{\mathrm{pa}(j)}\right)}{\int_{\mathcal{V}_i} p_j\left(v_j \mid \boldsymbol{v}_{\mathrm{pa}(j)}\right)p_i(v_i)\,\mathrm{d}v_i}\right) p(\boldsymbol{v})\,\mathrm{d}\boldsymbol{v}\,. \tag{C.95}$$

and similarly

$$\mathfrak{C}_{\pi(i)\to\pi(j)}^{Q\boldsymbol{z}} = \int_{\mathcal{Z}} \log\left(\frac{q_{\pi(j)}\left(z_{\pi(j)} \mid \boldsymbol{z}_{\mathrm{pa}(\pi(j);G')}\right)}{\int_{\mathcal{Z}_{\pi(i)}} q_{\pi(j)}\left(z_{\pi(j)} \mid \boldsymbol{z}_{\mathrm{pa}(\pi(j);G')}\right)q_{\pi(i)}(z_{\pi(i)})\,\mathrm{d}z_{\pi(i)}}\right) q(\boldsymbol{z})\,\mathrm{d}\boldsymbol{z}\,. \tag{C.96}$$

Since $\boldsymbol{Z} = \boldsymbol{P}_{\pi^{-1}} \circ \phi(\boldsymbol{V})$, we have $V_i = \psi_i(Z_{\pi(i)})$ for all $i \in [n]$ where $\psi = \phi^{-1}$.

Thus, by the change of variable formula, and using the fact that $\pi(\mathrm{pa}(i)) = \mathrm{pa}(\pi(i); G')$ since $\pi : G \mapsto G'$ is a graph isomorphism, we have for all $i \in [n]$:

$$q_{\pi(i)}\left(z_{\pi(i)} \mid \boldsymbol{z}_{\mathrm{pa}(\pi(i);G')}\right) = p_i\left(\psi_i\left(z_{\pi(i)}\right) \mid \psi_{\mathrm{pa}(i)}\left(\boldsymbol{z}_{\mathrm{pa}(\pi(i);G')}\right)\right)\left|\frac{\mathrm{d}\psi_i}{\mathrm{d}z_{\pi(i)}}\left(z_{\pi(i)}\right)\right|\,, \tag{C.97}$$

as well as for the marginal density

$$q_{\pi(i)}\left(z_{\pi(i)}\right) = p_i\left(\psi_i\left(z_{\pi(i)}\right)\right)\left|\frac{\mathrm{d}\psi_i}{\mathrm{d}z_{\pi(i)}}\left(z_{\pi(i)}\right)\right|\,, \tag{C.98}$$

and

$$q(\boldsymbol{z}) = p(\psi \circ \boldsymbol{P}_\pi(\boldsymbol{z}))\left|\det \boldsymbol{J}_\psi(\boldsymbol{z})\right|\,. \tag{C.99}$$

Substitution into the expression for $\mathfrak{C}_{\pi(i)\to\pi(j)}^{Q\boldsymbol{z}}$ yields:

$$\mathfrak{C}_{\pi(i)\to\pi(j)}^{Q\boldsymbol{z}} = \int_{\mathcal{Z}} \log\left(\frac{p_j\left(\psi_j(z_{\pi(j)}) \mid \psi_{\mathrm{pa}(j)}(\boldsymbol{z}_{\mathrm{pa}(\pi(j);G')})\right)}{\int_{\mathcal{Z}_{\pi(i)}} p_j\left(\psi_j(z_{\pi(j)}) \mid \psi_{\mathrm{pa}(j)}(\boldsymbol{z}_{\mathrm{pa}(\pi(j);G')})\right)p_i(\psi_i(z_{\pi(i)}))\left|\frac{\mathrm{d}\psi_i}{\mathrm{d}z_{\pi(i)}}(z_{\pi(i)})\right|\,\mathrm{d}z_{\pi(i)}}\right)$$

$$\tag{C.100}$$

$$p(\psi \circ \boldsymbol{P}_\pi(\boldsymbol{z}))\left|\det \boldsymbol{J}_\psi(\boldsymbol{z})\right|\,\mathrm{d}\boldsymbol{z}\,. \tag{C.101}$$

$$= \int_{\mathcal{V}} \log\left(\frac{p_j\left(v_j \mid \boldsymbol{v}_{\mathrm{pa}(j)}\right)}{\int_{\mathcal{V}_i} p_j\left(v_j \mid \boldsymbol{v}_{\mathrm{pa}(j)}\right)p_i(v_i)\,\mathrm{d}v_i}\right) p(\boldsymbol{v})\,\mathrm{d}\boldsymbol{v} \tag{C.102}$$

$$= \mathfrak{C}_{i\to j}^{P\boldsymbol{v}}\,. \tag{C.103}$$

where the second to last line follows by integration by substitution, applied to both integrals. $\qquad\square$

# D  Experimental Details and Additional Results

In this appemndix, we describe the experiments presented in § 6 in more details (Appx. D.1), and present additional results (Appx. D.2).

## D.1  Experimental Details for § 6

**Synthetic Data Generating Process.**  We consider linear Gaussian latent SCMs of the form

$$V_1 := U_1, \qquad V_2 := \alpha V_1 + U_2, \tag{D.1}$$

with standard normal $U_1$ and $U_2$. As a mixing function, we use a three-layer multilayer perceptron (MLP),

$$f = \sigma \circ \boldsymbol{W}_3 \circ \sigma \circ \boldsymbol{W}_2 \circ \sigma \circ \boldsymbol{W}_1 \tag{D.2}$$

where $\boldsymbol{W}_1, \boldsymbol{W}_2, \boldsymbol{W}_3 \in \mathbb{R}^{2 \times 2}$ are invertible weight matrices, and $\sigma$ is an element-wise invertible nonlinear leaky-tanh activation function used in [41]:

$$\sigma(x) = \tanh(x) + 0.1x. \tag{D.3}$$

To compute averages of our results over multiple runs, we construct different ground truth data generating processes as follows. We generate different latent SCMs by drawing $\alpha$ uniformly from $[-10, -2] \cup [2, 10]$. (We exclude $(-2, 2)$ to avoid sampling near unfaithful models.) We generate the corresponding mixing functions by uniformly sampling each element of the weight matrices, $(\boldsymbol{W}_k)_{ij} \sim U(0, 1)$. (To avoid the sampled weight matrices being too close to singular, we reject and resample if $|\det \boldsymbol{W}_k| < 0.1$.)

**Interventional Environments.**  In line with Thm. 3.2, for each choice of latent SCM and mixing function, we generate three environments: one observational environment and one interventional environment for each perfect single-node intervention. For $i = 1, 2$, we model a perfect intervention on $V_i$ by removing the influence of the parent variables and changing the exogenous noise by shifting its mean up or down. Specifically, we replace the corresponding assignment in (D.1) by

$$V_i := \tilde{U}_i, \quad \text{where} \quad \tilde{U}_i \sim \mathcal{N}(m_i, 1) \tag{D.4}$$

where the mean $m_i$ of the shifted Gaussian noise is fixed per environment and sampled uniformly from $\{\pm 2\}$.

We label the observational environment as $e = 0$ and the environment arising from intervention on $V_i$ by $e = i$ for $i = 1, 2$. Samples from $p^e$ are then generated by sampling latents $v$ from the respective (un)intervened SCM and then applying the mixing function.

**Model Architecture.**  We use normalizing flows [93] to model observations $x$ as the result of an invertible, differentiable transformation $g$ of some latent (noise) variable $z$,

$$x = g(z). \tag{D.5}$$

We apply a series of $L$ such transformations $g^l : \mathbb{R}^2 \to \mathbb{R}^2$ such that $g = g^L \circ \ldots \circ g^1$ which we refer to as *flow layers*. We use Neural Spline Flows [30] for the invertible transformation, with a 3-layer feedforward neural network with hidden dimension 128 and permutation in each flow layer and $L = 12$ layers. The transformations $g, g^1, \ldots, g^L$ have learnable parameters (the weights and biases of the neural networks), which we omit to simplify notation.

Typically, simple distributions such as a uniform or isotropic Gaussian are used as base distribution $q(z)$ in normalizing flows. Here, we instead choose a base distribution that encodes information about the latent SCM. Specifically, we model the base mechanism as

$$q_1(z_1) = \mathcal{N}\left(\mu_1, \sigma_1^2\right), \qquad q_2(z_2 \mid z_1) = \mathcal{N}\left(\hat{\alpha} z_1, \sigma_2^2\right), \qquad q_2(z_2) = \mathcal{N}\left(\mu_2, \hat{\sigma}_2^2\right) \tag{D.6}$$

and the intervened mechanism as

$$\tilde{q}_1(z_1) = \mathcal{N}(\tilde{\mu}_1, \tilde{\sigma}_1^2), \qquad \tilde{q}_2(z_2) = \mathcal{N}(\tilde{\mu}_2, \tilde{\sigma}_2^2). \tag{D.7}$$

**Candidate Graphs and Intervention Targets.** We train a separate normalizing-flow based model for each choice of candidate graph $G'$ and inferred intervention targets. For the bivariate case with $n = 2$, this gives rise to four models, depending on whether $G'$ matches $G$ or not, and whether the intervention targets are aligned or misaligned w.r.t. the ground truth intervention targets. To model the setting $G' \neq G$ in which $Z_1$ and $Z_2$ are assumed independent, we use $q_2(z_2)$ in place of $q_2(z_2 \mid z_1)$ in (D.6). If the intervention targets are aligned, we use $\tilde{q}_i$ instead of $q_i$ in $e = i$ for $i = 1, 2$. Else, if they are misaligned, we use $\tilde{q}_2$ instead of $q_2$ in $e = 1$ and $\tilde{q}_1$ instead of $q_1$ in $e = 2$. By multiplying the respective mechanisms, we thus obtain three environment-specific joint base distributions $q^e(\boldsymbol{z})$ for $e = 0, 1, 2$.

**Learning Objective.** Given multi-environment data, the parameters $\mu_1, \sigma_1, \hat{\alpha}, \sigma_2, \mu_2, \hat{\sigma}_2, \tilde{\mu}_2, \tilde{\sigma}_2$, $\tilde{\mu}_1$ and $\tilde{\sigma}_1$ are jointly learned with the parameters of the invertible transformations $g^l$ by maximising the log-likelihood of the data under our model, which is given by:

$$\sum_{e \in \mathcal{E}} \mathbb{E}_{\boldsymbol{x} \sim p^e(\boldsymbol{x})} \left[ \log p^e_{\text{model}}(\boldsymbol{x}) \right] = \sum_{e \in \mathcal{E}} \mathbb{E}_{\boldsymbol{x} \sim p^e(\boldsymbol{x})} \left[ \log q^e(h(\boldsymbol{x})) + \log \left| \det \boldsymbol{J}_h(\boldsymbol{x}) \right| \right] \quad \text{(D.8)}$$

where the encoder $h := g^{-1}$ is the inverse of the normalizing flow which is readily available by construction; and where the expectations are empirical averages over the respective datasets in practice.

**Training and Model Selection Details.** Each environment comprises a total of 200k data points. We use the ADAM optimizer [67] with cosine annealing learning rate scheduling, starting with a learning rate of $5 \times 10^{-3}$ and ending with $1 \times 10^{-7}$. We train the model for 200 epochs with a batch size of 4096. We split the dataset into 70% for training, and 15% for validation and held-out test data, each sampled randomly across all environments. For each drawn data generating process, we train three versions of each model with different random initializations and select the one with the highest validation log likelihood at the end of training for evaluation.

**Evaluation Metrics.** We evaluate the trained models w.r.t. *mean correlation coefficient* (MCC) on held-out data and *log-likelihood* on validation data (for model selection).

- The MCC measures the extent to which there is a one-to-one correspondence between the ground truth latents $V_i$ and (a permuted version of) the inferred latents $Z_i = h_i(\boldsymbol{X})$. Its maximum value of one indicates a perfect correlation between the two. MCC is thus a proxy measure for the level of identifiability up to permutation and invertible reparametrisation. We report MCC based on Pearson (linear) correlation, though we found the results based on Spearman (nonlinear monotonic) correlation to be almost identical.

- The log-likelihood, on the other hand, measures how well a model explains or fits the data. Since the ground truth is typically unknown, a reasonable procedure when training multiple models is to select the one that attains the highest likelihood. For this reason, we report the difference in log-likelihood between misspecified models (ones assuming a wrong graph or intervention targets) to the correctly specified model. Whenever this difference is larger than zero, the correct model fits the data better and would thus be selected.

### D.2  Additional Results: Learning Nonlinear Latent SCMs from Partial Causal Order

In this subsection, we present an additional experiment, in which we extend the setting investigated in § 6 and Appx. D.1 along the following axes.

- We fit generative models over three instead of two variables, corresponding to the setting of Thm. 3.4.
- The ground-truth SCM is now given by nonlinear mechanisms with non-additive, non-Gaussian noise.
- The generative model, including the learnt mechanisms, is now fully nonlinear.
- Despite Thm. 3.4 formally requiring two environments per single-node intervention, we only provide one interventional environment per node.
- Rather than searching over candidate graphs, we only fix the causal order and fit the reduced form of the SCM (see § 2.1) with a second normalizing flow.

Below, we describe these differences in more detail.

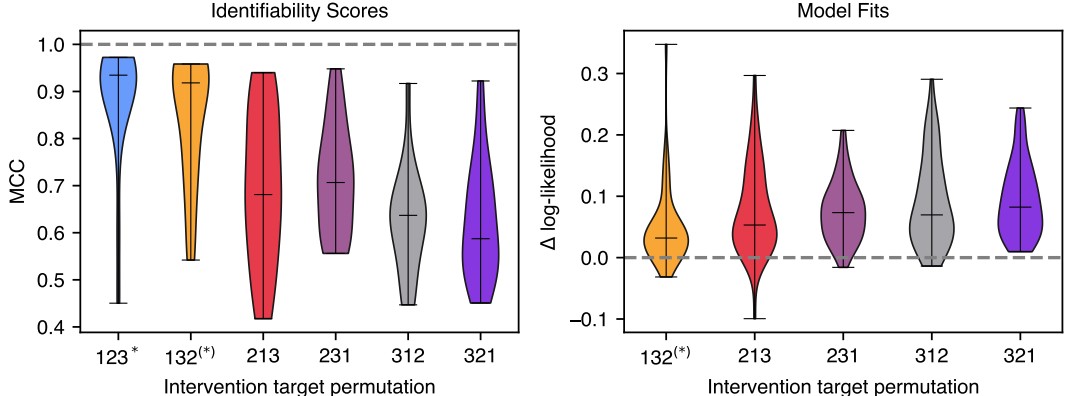

Figure 4: **Comparison of Correctly and Incorrectly Specified Models for $V_2 \leftarrow V_1 \rightarrow V_3$ with Fixed Causal Order and Nonlinear SCM.** Each violinplot corresponds to one setting where the intervention target labels are permuted. The blue plot ($123^*$) is the setting with correct intervention target labels. The yellow plot ($132^{(*)}$) has the targets for the two children $V_2$ and $V_3$ permuted, which also corresponds to a correct causal ordering and should thus be considered equivalent. We show mean correlation coefficients (MCCs) between the learned and ground truth latents *(Left)* and the difference in validation model log-likelihood between the well-specified (blue) and misspecified models *(Right)*. Each violin plot is based on 20 different ground truth data generating processes; the horizontal lines indicate the minimum, median and maximum values.

**Three-Variable Graph.** The *unknown* ground truth graph is given by

$$V_2 \leftarrow V_1 \rightarrow V_3\,. \tag{D.9}$$

This is consistent with the partial ordering $V_1 \preceq V_2 \preceq V_3$, which is assumed for all models a priori w.l.o.g., see § 2.2. Note that, due to the encoding of causal structure in the nonparametric model explained below, we only iterate over different permutations of the intervention targets and not over latent graph configurations. Due to the causal order implied by the graph (D.9), the permutations $(1, 2, 3)$ (no permutaion) and $(1, 3, 2)$ (permutation of the two effects) are equivalent since the latter also implies the correct causal ordering.

**Nonlinear, Non-Gaussian SCM.** The mechanisms in the ground-truth SCM are now given by

$$V_i := \beta f_i^{\mathrm{loc}}(\boldsymbol{V}_{\mathrm{pa}(i)}) + f_i^{\mathrm{scale}}(\boldsymbol{V}_{\mathrm{pa}(i)})U_i \tag{D.10}$$

for all $i$, where the location and scale functions $f_i^{\mathrm{loc}}, f_i^{\mathrm{scale}} : \mathbb{R}^{|\mathrm{pa}(i)|} \to \mathbb{R}$ are parameterized by random 3-layer neural networks (sampled as the random mixing function in (D.2)) and the noise variables are Gaussian, $U_i \sim \mathcal{N}(0, 1)$. The factor $\beta$ controls the influence of the parent variables relative to the exogenous noise. As $\beta$ increases, variables tend to become more dependent, as also the mean shifts as a function of the parent variables. We set $\beta = 10$ for the experiments shown in Fig. 4.

**Nonparametric Latent SCM.** We use a second normalizing flow to learn a *reduced form of the latent SCM* via the transformation $g^{\mathrm{SCM}} : \mathbb{R}^3 \to \mathbb{R}^3$ mapping an exogenous noise variable $\boldsymbol{\epsilon}$ to the latent variable $\boldsymbol{z}$,

$$\boldsymbol{z} = g^{\mathrm{SCM}}(\boldsymbol{\epsilon})\,. \tag{D.11}$$

The distribution of the exogenous noise variable $\boldsymbol{\epsilon}$ as well as the distribution of the intervened mechanisms $\tilde{q}_i(z_i)$ for $i = 1, 2, 3$ is fixed and standard (isotropic) Gaussian. The flow layers in $g^{\mathrm{SCM}}$ have an *upper triangular Jacobian* and thus allow us to encode assumptions about the causal graph: by passing the variables in topological order, which we can assume w.l.o.g., we ensure that an exogenous noise variable $\epsilon_i$ can only influence endogenous variables in $\boldsymbol{z}$ that are descendants of $z_i$. The learned weights of the flow layers then implicitly encode which endogenous variables are connected. Therefore, only different choices of the permutations of the intervention targets need to be considered as candidate models. We use a similar architecture based on Neural Spline Flows. However, we omit permutation layers, which would violate the topological order of the variables.

**Results.** In Fig. 4, we present identifiability scores and model fits for both well-specified and misspecified models (corresponding to different intervention target choices). Notably, we observe that the well-specified model (in blue) or its equivalent (in yellow) yield the highest log-likelihood in the majority of cases, as depicted in Fig. 4 (*Right*). This demonstrates that, even in this nonparametric setting without fully specified graph, the log-likelihood remains a reliable criterion for selecting the correct intervention targets. Fig. 4 (*Left*) shows that the selected models (blue or yellow) approximately identify the ground-truth latent variables up to element-wise rescaling, whereas other choices lead to much lower MCCs.

It is worth noting that, compared to the parametric setting investigated in § 6 and Fig. 3, the nonparametric setting appears to be more challenging (as expected), as there is a less pronounced distinction between well-specified and misspecified models, both in terms of identifiability scores and model fits. Moreover, future work is needed to parse the implicitly learned causal relationships in the transformation $g^{\mathrm{SCM}}$ in (D.11): since only the (pre-imposed) causal order is specified, in practice, $g^{\mathrm{SCM}}$ may learn to use additional or fewer edges than in the true graph $G$.

# E  Discussion of the Role of Our Assumptions

Below, we summarize the rationale and intuition behind each assumption:

- Asm. 2.2 helps rule out degenerate cases (cancellation along different paths) in which variables are (conditionally) independent despite being causally related. It is a standard assumption in classical causal discovery from observational data, and therefore also helps in CRL to recover the true causal graph.

- Asm. 2.3 is required to know how many latent variables we are looking for. It is a standard assumption in identifiable representation learning (that is often made implicitly). However, it may be dropped when suitable techniques for estimating the intrinsic dimensionality of $\mathcal{X}$ can be employed.

- Asm. 2.5 is needed for the mapping between latents and observations to be invertible in the first place. Without it, full recovery of the causal variables (up to CRL equivalence) is infeasible. This assumption is also standard for the simpler problem of nonlinear ICA.

- Asm. 2.8 is a characterisation of our generative setup. Sharing of some mechanisms and the mixing function is needed for the multi-environment setting to provide useful additional information: if everything may change across environments, the datasets can only be analysed in isolation, running into the non-identifiability of CRL from iid data.

- Asm. 2.9 and (A2) / (A2') are needed since with imperfect interventions or interventions not on all nodes, identifiability is not achievable even in the linear setting as shown by Squires et al. [117].

- Asm. (A1) is a technical assumption needed for our analysis. It is not strictly necessary (it can also be relaxed to fully supported on a Cartesian product of intervals) but substantially eases the readability and accessibility of the proof, without a major impact on the main causal aspects of the problem setup.

- Asm. (A3) / (A3') is needed to avoid spurious solutions based on applying a measure preserving transformation on a part of the domain unaffected by the intervention.

- Asm. (A4) is needed to rule out a fine-tuning of the ground-truth generating process that are possible due to fully non-parametric nature of the setup, see also Remark 4.2 and the following paragraph.

