# OpenReview forum: "Nonparametric Identifiability of Causal Representations from Unknown Interventions"
_NeurIPS.cc/2023/Conference — NeurIPS 2023 poster_

### Official Review · Reviewer_ZLAN · 2023-06-29

**Soundness:** 3 good
**Presentation:** 3 good
**Contribution:** 3 good
**Rating:** 6
**Confidence:** 4

**Summary:**

The paper discusses the task of identifying causal variables from high-dimensional observations under non-parametric mixing functions and causal mechanisms. This is done under the assumption of single-node, perfect interventions being available for all causal variables, as well as distinct paired perfect interventions in the case of having more more than two causal variables. The paper proves that causal variables are identifiable under this setup, when taking additional assumptions on the interventions being sufficiently different from the observational distribution. Thereby, weaker assumptions are possible for 2 variables than for more variables. Finally, the paper sketches possible implementations of learning algorithms for this setting.

**Strengths:**

The paper is overall well written, even if it is aimed at researchers in identifiability and/or causal representation learning (CRL) specifically. A consistent notation is used throughout the paper, and all assumptions are clearly stated before the theorems. It is appreciated that proof sketches have been included in the main paper to support the claimed theorems and make the main paper a bit more standalone. The paper discusses all necessary related work and puts itself into context of the current field of research.

The main contribution of the paper is its theoretical result. It extends the domain of identifiable causal representation by considering yet another setup, where environment pairs with single-node, perfect interventions are given. The benefit of this setup is that it does not require counterfactual observations, while supporting a large function class despite taking needed assumptions on the interventions. The proofs for supporting the claimed theorems are given in the appendix, following common proof strategies in CRL. The proofs appear sound and intuitive, although a very careful check of the proofs was not possible during the review period. Overall, it is a good contribution to the theoretical identifiability in CRL.

**Weaknesses:**

While the derived theory in the paper puts weaker constraints on the mechanisms of the causal variables and mixing function, its assumption of having access to single-node, perfect interventions on all causal variables is restrictive. Being able to perform an intervention on a variable is already commonly considered expensive or often not easily feasible, especially if it is a perfect intervention and single-node. However, doing this twice and even different between the two setups is challenging. Further, obtaining such a dataset requires non-trivial prior knowledge of the causal system, since it necessitates the ability to perform such single-node, perfect interventions on causal variables that are yet to be identified. The paper misses to give real-world examples to motivate the setup and its assumptions, which puts it in a more limited spot.

Besides the theoretical results, it is also important to validate the setup and the practicality of the theory in empirical studies. The paper only sketches some potential ideas, where all unknown parts are learned. However, optimizing the latent encoder, the causal graph, and the intervention targets all at the same time is not trivial as shown in previous works. Further, the appendix shows some limited results on a generative model, where one would need to iterate over all possible causal graphs and intervention targets. Still, this is not practical for systems larger than very few causal variables or high-dimensional observations.

The paper states that the intervention targets are not known. However, under the identifiability up to permutation, the intervention targets in this setup appear to be known. Specifically, assumption (A2') states that there exist $n$ environment pairs, where each pair intervenes on a different causal variable. Thus, the intervention targets for these pairs, as stated in the assumption, are known as $\pi(i)$. Since the variables cannot be identified up to permutation $\pi$ anyway, permuting the causal variables and thus the targets are still considered to be the same targets in the same identifiability class, e.g. as in the works cited for known intervention targets [69, 70]. Thus, the claim of unknown intervention targets appears not valid given the assumptions, or the assumptions should be clarified to e.g. have at least $n$/$n+1$ environments.

### Typos:
- Table 1: 'Causal Representation Learning'

**Questions:**

### Review summary

The theoretical results of the paper provide a new setting under which causal variables are identifiable in CRL. However, the paper is limited by its strong reliance on single-node, perfect interventions and very limited empirical study. I consider the theoretical results outweighing the drawbacks a bit, although the paper would strongly benefit from empirical validation of the setup. Thus, my recommendation is 'Weak Accept'.

### Questions

- What is a real-world scenario in which the setup of pairs of single-node, perfect interventions is practical and common?
- Do you require the knowledge of the intervention targets up to permutation, or do you allow for more environments/environment pairs as long as each variable has been intervened upon once?

**Limitations:**

Limitations have been discussed in different parts of the paper.

---

> ### Author Rebuttal · Authors · 2023-08-09
>
> Thank you very much for reviewing our work. We address your questions and comments below.
> ___
>
> > Restrictiveness of the assumption of access to all single-node perfect interventions.
>
>
> While we agree that this is a strong assumption, it has previously been shown to be necessary even for more restricted settings. For example, Squires et al. [111] showed the necessity of access to all single node perfect interventions even in the fully linear interventional case, and Brehmer et al. [18] did so for a counterfactual, multi-view nonparametric setting. In this sense, it is unsurprising that at least as strong an assumption is needed in the nonparametric interventional case (and interesting that the same can be sufficient).
> ___
>
> > Limited empirical study and the need to iterate over all possible causal graphs
>
> As described in more detail in our general response to all reviewers, we now provided additional empirical results for a nonparametric setting with three variables. In particular, enumeration of causal graphs is avoided in this setting. Our main focus was on new theoretical results and validations thereof, and we believe that our new and existing experimental results are helpful to this end. That being said, we completely agree that “optimizing the latent encoder, the causal graph, and the intervention targets all at the same time is not trivial” and that further work is needed to make CRL methods more practical.
> ___
>
> > Clarification regarding assumption (A2’) and (un)known intervention targets.
>
> This is a subtle point that may not have been sufficiently clear in the initial manuscript. For the setting of Thm 4.2, assumption (A2’) states that (i) datasets come in pairs; (ii) we know that, within each pair, the same variable was intervened upon; (iii) we do not know which variable was intervened upon in each pair, only that each variable was intervened upon (at least) in one pair. Importantly, note that $\pi$ in (A2’) is defined as any permutation of $[n]$ and is not restricted to the subset of permutations that are isomorphisms of the unknown true graph (which is the partial permutation ambiguity of the CRL equivalence class). The conclusion that this is the same as knowing the intervention targets is therefore not supported. We will make the above more clear in the revised manuscript.
>
> We are not quite sure what exactly is meant by “the assumptions should be clarified to e.g. have at least n/n+1 environments” and would welcome further clarification if this remains an open issue after our response.
>
> ___
>
> > Real world examples
>
> We are not experts in this domain (so the following is to be taken with a grain of salt), but we try to sketch a possible scenario and how it relates to our assumptions.
>
> Consider genetic experiments, where tools like CRISPR-Cas9 may permit one to perform targeted and (near) perfect interventions on individual genes. The observations in this case could correspond to some downstream effect that is influenced by gene activity, such as counts of different proteins. One might target different genes (for multiple cells) and thus collect multiple environments or datasets. Moreover, different types or concentrations of CRISPR-Cas9 might give rise to (two or more) interventional environments affecting the same target. It may be natural to think of this setting as one with known intervention targets. However, if we consider a less well-studied genome of another species, the exact places where certain DNA or RNA sequences are found may not be known a prior, corresponding to unknown intervention targets.
>
> We will happily add the above example, or other suggestions the reviewer may have, to the revised manuscript.

---

> > ### Comment · Reviewer_ZLAN · 2023-08-11
> > **Response to Rebuttal**
> >
> > Thank you for your response and clarifications.
> >
> > > Clarification regarding assumption (A2’) and (un)known intervention targets.
> >
> > I generally agree with you that your case indeed does not require knowing the intervention targets. My comment was mostly regarding the assumption A2, which currently suggests that this would be the case. By replacing the phrase of "*there exist $n+1$ environments...*" to "*there exist at least $n+1$ environments...*" (line 240), this could be clarified. This is because if you have exactly $n+1$ environments, the intervention targets can be inferred (under some definitions of previous works), and if you have more than $n+1$, the intervention target cannot be inferred anymore. If we take the example of three variables given in the general response, having four datasets, with the first being the observational one, would implicitly give you the intervention targets by numbering the dataset pairs in any arbitrary order. No restriction to isomorphisms of the unknown graph are needed, if one learns an arbitrary graph. If you have more than $n+1$ datasets, this is not possible anymore.
> >
> > > Emperical results
> >
> > It is good to see additional results. Still, the setting and usability of the method is signifciantly limited given the iteration over intervention targets. This requires training up to $n!$ models, a number that can quickly go out of hands for $n>3$ and slightly more expensive datasets to train on. Further, given the small differences and high standard deviations (e.g. diff between 132 and 231 despite inverting the graph), it is unlikely that the empirical method is applicable in challenging datasets at the moment.
> >
> > Thus, the empirical part remains a considerable weakness of the paper. Nonetheless, as mentioned in my original review, the strengths from the theory outweigh this, in particular when the paper is improved by the suggestions of the reviewers.

---

> > > ### Author Response · Authors · 2023-08-18
> > > **Author Response**
> > >
> > > Thank you for the clarification, continued engagement, and insightful remarks.
> > > _____
> > >
> > > **Known vs Unknown Targets.** What exactly constitutes known vs unknown intervention targets appears to be a less clearly defined concept in CRL than in the fully observed case.
> > >
> > > In principle, we agree that *if all possible causal graphs are considered* and exactly $n$ interventional environments with distinct targets are provided, then one can indeed simply call variable $V_k$ the one intervened upon in environment $k$ for $k=1, …, n$, and consider the intervention targets *known in this sense*.
> > >
> > > In our work, we instead consider a setup in which the causal ordering is fixed to $V_1 \preceq V_2 \preceq … \preceq V_n$ and only graphs consistent with this ordering are considered (see l.167 ff.). This formulation is motivated by starting from the generative process and its fundamental ambiguities (e.g., ordering of the nodes), comes w.l.o.g., and was also adopted in some recent works, see, e.g., Squires et al. [111, Remark 1]. The intervention targets are then considered *unknown w.r.t. this pre-imposed causal ordering*. Our current proofs show that the intervention targets can then be identified from exactly $n$ environments (possibly up to irresolvable partial re-ordering), even if they are not known (w.r.t. the pre-imposed causal order) a priori.
> > > ______
> > >
> > > **More than $n$ environments.** Regarding the “at least” formulation of allowing $m>n$ interventional environments, this would indeed strengthen the results and break the suggested strategy of assigning target $k$ to interventional environment $k$. In this case, we do *not* know a suitable subset of $n$ environments which contains exactly one intervention for each node. It therefore first needs to be shown that such a subset of environments can be identified.
> > >
> > > Suppose for a contradiction that we select a subset of $n$ interventional environments which are assumed to correspond to distinct targets in the model $Q$ whereas this is not the case for the ground truth $P$ (i.e., there are actually duplicate and missing interventions).
> > >
> > > - For the setting of Thm. 4.3 with paired interventions, we can show that this is not possible: Suppose that there are two pairs of environments $(e_a, e_a’)$ and $(e_b, e_b’)$ corresponding to interventions on $V_i$ in $P$, but which are modelled as interventions on distinct nodes $Z_j$ and $Z_k$ in $Q$. Similar to the proof sketch in l.301-303, it can be shown that $V_i$ must then simultaneously be a deterministic function of only $Z_j$ and only $Z_k$. This implies that $\partial \psi_i / \partial z_l =0$ for all $l$ which contradicts invertibility of $\psi$. Hence, only valid subsets will not lead to a contradiction. We can thus allow for any number $m\geq n$ of paired environments (“completely unknown targets”) for Thm. 4.3, as long as there is at least one paired intervention for each node. We will adjust assumption (A2’) to reflect this generalization, and will add a more detailed version of the above argument to the proof.
> > >
> > > - For the setting of Thm. 4.1 with single interventions, finding a contradiction unfortunately seems more difficult. In short, we can rule out duplicate interventions on root nodes, but for $V_1 \to V_2$ we were not yet able to find a contradiction to selecting a subset of environments corresponding to two interventions on $V_2$. We will continue to investigate this matter, but remark that prior results in simpler parametric settings (e.g., Squires et al. [111], Varici et al. [116]) also require access to a set of exactly $n$ interventional environments, one for each node.
> > >
> > > We will add a summary paragraph about the subtleties of known vs. unknown intervention targets in CRL to the discussion.

---

### Official Review · Reviewer_E44d · 2023-07-03

**Soundness:** 3 good
**Presentation:** 3 good
**Contribution:** 3 good
**Rating:** 6
**Confidence:** 2

**Summary:**

The paper studies the problem of inferring causal relationships between $n$ latent variables through observations under a mixing function. Given data $X$ from multiple environments (each of which corresponds to an unknown perfect atomic intervention), where $X$ is the observation of the latents under a fixed mixing function $f$, the goal is to recover $f^{-1}$ and the causal graph $G$ on the latent variables (up to $\sim_{CRL}$ equivalence).

**Strengths:**

The problem is well-motivated and interesting. The authors did a good job explaining how this paper differs from prior work while providing a pretty good literature review. Some experiments are also given in Appendix D.

**Weaknesses:**

While I am not an expert in the area and did not check all the proofs in detail, I do not see any glaring weaknesses. The theorem statements and proof sketches seem believable, especially since there is a lot of assumptions that were made to "make things go through". My biggest gripe is that there is a lack of discussion about the assumptions (see Questions section).

**Questions:**

Line 175:
Maybe write what CRL stands for somewhere (possibly in the footnotes)?

Assumptions:
There are a lot of assumptions (which is okay, if they are well-justified and discussed). Can you explain or discuss why each of them is necessary or reasonable to have (without trivializing the problem)? I understand that you believe "pairs of environments" is not necessary in general, but what about the other assumptions? What happens if all but one is satisfied? What goes wrong? I am happy to further increase my score if this is sufficiently addressed and if the other reviewers did not raise any damning issues that I missed.

Table 1 caption:
Typo: "Reresentation Learning"

**Limitations:**

Nil.

---

> ### Author Rebuttal · Authors · 2023-08-09
>
> Thank you very much for reviewing our work. We will fix the typo and add an explanation for the CRL abbreviation. We answer your main question below.
>
> > There are a lot of assumptions (which is okay, if they are well-justified and discussed). Can you explain or discuss why each of them is necessary or reasonable to have (without trivializing the problem)?
>
> We summarize below the rationale behind each assumption, most of which are also highlighted in our proof sketches.
> - Asm. 3.2 is required to rule out degenerate cases (cancellation along different paths) in which variables are (conditionally) independent despite being causally related. It is a standard assumption in classical causal discovery, and therefore also needed in CRL to recover the causal graph.
> - Asm. 3.3 is required to know how many latent variables we are looking for. It is a standard assumption in identifiable representation learning (that is often made implicitly). However, as discussed in footnote 8 and our response to reviewer `U76e`, it may be dropped when suitable techniques for estimating the intrinsic dimensionality of $\mathcal{X}$ can be employed.
> - Asm. 3.5 is needed for the mapping between latents and observations to be invertible in the first place. Without it, full recovery of the causal variables (up to CRL equivalence) is infeasible. This assumption is also standard for the simpler problem of nonlinear ICA.
> - Asm. 3.9 is a characterisation of our generative setup. Sharing of some mechanisms and the mixing function is needed for the multi-environment setting to provide useful additional information: if everything may change across environments, the datasets can only be analysed in isolation, running into the non-identifiability of CRL from iid data.
> - Asm. 3.10 and (A2) / (A2’) are needed since with imperfect interventions or interventions not on all nodes, identifiability is not achievable even in the linear setting as shown by Squires et al. [104].
> - Asm. (A1) is a technical assumption needed for our analysis. It is not strictly necessary (see, e.g., footnote 9 on p. 21 in Appx. C.3) but substantially eases the readability and accessibility of the proof, without a major impact on the main causal aspects on the problem setup.
> - Asm. (A3) / (A3’) is needed to avoid spurious solutions based on applying a measure preserving transformation on a part of the domain unaffected by the intervention, see also l. 318-320.
> - Asm. (A4) is needed to rule out a fine-tuning of the ground-truth generating process that are possible due to fully non-parametric nature of the setup, see also Remark 4.2 and the following paragraph.
>
> We hope that this sufficiently addresses your question. If so, we kindly ask whether you would consider increasing your score, also considering the lack of “any damning issues raised by other reviewers”.

---

> > ### Comment · Reviewer_E44d · 2023-08-11
> >
> > Thank you for addressing my concerns. Please add a version of the above discussions about the assumptions in the revision.
> >
> > I have increased my score :)

---

### Official Review · Reviewer_U76e · 2023-07-05

**Soundness:** 3 good
**Presentation:** 4 excellent
**Contribution:** 3 good
**Rating:** 7
**Confidence:** 3

**Summary:**

This paper proposed to identify the latent causal representations and their underlying causal structure, which is a very challenging and interesting problem. The \sim_{CRL} is introduced to describe the equivalent class up to elementwise operations and permutation, which is sufficiently meaningful for practical use. The CRL-identifiability theory is given under the data from paired interventional data and other assumptions, such as the pre-given number of nodes and others. In appendix, the authors presented a simple version of learning method and validates it on a synthetic dataset.

**Strengths:**

In general, I found this paper to be highly enjoyable and insightful. It successfully addresses a challenging and captivating problem of extracting causal representations and their relationships. Given the increasing prevalence of unstructured data, such an endeavor holds significant importance. The authors have provided a comprehensive overview and engaging discussions that effectively highlight the unique contributions of their work in relation to existing literature. Moreover, the use of paired interventional data, which is more readily obtainable in practical scenarios, adds to the paper's practical relevance. Besides, the organization and writing of this paper are commendable.

**Weaknesses:**

1. I recommend that the authors provide practical demonstrations of the proposed method in real-world scenarios. While acquiring paired interventional data can be challenging in real-world settings, the authors could consider utilizing datasets generated from virtual environments, such as the causal world (https://sites.google.com/view/causal-world/home), to showcase the utility of their approach.

2. In practical applications, determining the number of latent nodes n, is often difficult. Consequently, verifying whether the number of paired intervention data includes all latent variables becomes challenging. This limitation may restrict the scope of application for the proposed theory and learning methods.

**Questions:**

1. Intuitively, it appears that with paired interventional data, we can identify the latent representation with a unique permutation. For example, if we intervene to place a ball at position A on the table in e_i and at position B on the table in e'_i, we can infer that the position is the intervened variable through a simple comparison. Could you please provide a more in-depth explanation of the challenges encountered in theoretical analysis?

2. The requirement of the genericity condition is specified in Theorem 4.1, whereas it is not explicitly mentioned in Theorem 4.2, which is presented as a more general version of Theorem 4.1. Additionally, the assumption A_2' does not degenerate to A_2 when n=2; instead, it is stronger than extending A_2 to a general n. I would appreciate a more detailed clarification regarding this matter.


**Limitations:**

Yes, the authors adequately addressed the limitations.

---

> ### Author Rebuttal · Authors · 2023-08-09
>
> Thank you very much for reviewing our work. We address your two main questions and other comments below.
>
> ### Main questions
>
> > “Intuitively, it appears that with paired interventional data, we can identify the latent representation with a unique permutation. For example, if we intervene to place a ball at position A on the table in e_i and at position B on the table in e'_i, we can infer that the position is the intervened variable through a simple comparison. Could you please provide a more in-depth explanation of the challenges encountered in theoretical analysis?”
>
> If we correctly interpret the question, there may be a misunderstanding as to what we mean by “pairs of environments” (“paired interventional data” in your comment). Unlike other works considering counterfactual, multi-view data (e.g., [17,110]), we do NOT observe pairs of observations $(x,x’)$ as in your example with the moving ball. Instead, we only have access to two datasets drawn from $P^1_X$ and $P^2_X$ which we know both arise from two distinct stochastic interventions on the same *unknown* latent variable. However, none of the images in the two datasets need to show the same ball (with only position differing): all other latent variables are resampled for each observation and will never agree exactly across datasets (except on a set of measure zero). At a high level, the main challenge of the analysis (compared to a counterfactual or multi-view setting) thus stems from the lack of correspondence across observations from different datasets: we may never see the same object more than once, and it may be non-trivial to infer what was intervened upon if multiple generative factors differ.
>
> Moreover, in your example, we only observe the pixels but not directly the position variable; the mapping between latents and observations, as well as the latent distributions are unknown and may be arbitrarily complex due to the nonparametric nature of the problem. Another key challenge thus arises from the flexible nonparametric problem setting.
>
> Please let us know if we misunderstood your question or require further clarification.
>
> ___
>
> > The requirement of the genericity condition is specified in Theorem 4.1, whereas it is not explicitly mentioned in Theorem 4.2, which is presented as a more general version of Theorem 4.1. Additionally, the assumption A_2' does not degenerate to A_2 when n=2; instead, it is stronger than extending A_2 to a general n. I would appreciate a more detailed clarification regarding this matter.
>
> The distinction between the two settings may not have been emphasised enough in the initial manuscript. Allow us to clarify: Thm. 4.2 is not simply a direct extension of Thm. 4.1 to $n>2$, but a different statement relying on a related but distinct set of assumptions. As you correctly note, assumption (A2’) in Thm. 4.2 is indeed stronger than and different from assumption (A2) in Thm. 4.1. We chose the similar naming as both are assumptions about the available environments (but we are open to changing this to prevent confusion). Due to the stronger assumption (A2’), the genericity condition is indeed not needed for Thm. 4.2. We hope that this clarifies the matter and will highlight these subtleties in the revised manuscript.
>
> ___
>
> ### Other comments
>
> > “practical demonstrations of the proposed method in real-world scenarios”
>
> While we appreciate the suggestion and agree that this would be interesting, this was unfortunately out of the scope for the limited rebuttal period.
>
> However, we refer to Appendix D and our general response to all reviewers for more details on our experiments (on synthetic data), parts of which we will move to the main text.
>
> > “determining the number of latent nodes $n$”
>
> We agree that this can be challenging, which is also why we explicitly included Asm. 3.3. We briefly touch upon practical methods for estimating the dimensionality $n$ of the observational manifold in footnote 8 in the last paragraph, which we will happily move to the main text.

---

> > ### Comment · Reviewer_U76e · 2023-08-19
> >
> > Thank you for addressing my concerns. I keep my original score.

---

### Official Review · Reviewer_FsUF · 2023-07-10

**Soundness:** 3 good
**Presentation:** 4 excellent
**Contribution:** 3 good
**Rating:** 6
**Confidence:** 3

**Summary:**

This paper gives an identifiability result in a setting that is relevant to causal representation learning, where we wish to infer latent causal variables and their causal graph from high-dimensional observations. They work in a setting that is more general than prior work that relies on, for example, weak supervision, temporal structure, or known intervention targets. Their setting assumes that both the causal model and the mixing function are nonparametric, and the targets of the interventions are unknown.

Their identifiability results are up to trivial indeterminacies (permutations and element-wise diffeomorphisms) and identify both the causal graph and the mixing function. Their first theoretical result shows identifiability for two causal variables given one perfect stochastic intervention per node. Their second theoretical result shows identifiability for an arbitrary number of variables when there are two paired perfect stochastic interventions per node.

The main text of the paper does not have an experiment section.

**Strengths:**

- This paper frames a problem setting for identifiability that is interesting to causal representation learning, which begins to bridge the gap from existing identifiability results to modern machine learning that occurs on high-dimensional observed data.
- They work in a highly general setting where both the causal model and the mixing function are nonparametric, and the targets of the interventions are unknown. In my opinion, the problem framing and the choice of this general setting are the primary contributions of this work even if the theoretical results have limitations.
- The paper is generally well-written and well-structured.

**Weaknesses:**

- Their first theoretical result is in a setting with only two causal variables, where you have an observation distribution and one perfect intervention per node. This result would be much stronger in a setting with n>2, as the authors note in the conclusion.
- Their second theoretical result is in a setting with arbitrary number of variables, but requires two distinct perfect paired interventions. Requiring these pairs of interventions is not a terribly realistic assumption, even though they don't require the intervention targets to be known.
- There is no estimation method or experiment results. Other identifiability papers often contribute an estimation method (e.g. a VAE using a regularizer that encourages sparsity of a mixing function), perform disentanglement experiments in settings that match their theoretical assumptions, or perform ablations on synthetic data where they can control which of their theoretical assumptions are met in order to empirically study the necessity / sufficiency of their assumptions.

**Questions:**

- Does your theory suggest any empirical validation that you could add to this paper? See the last bullet point in "weaknesses" section for empirical approaches that could be relevant to this kind of theoretical contribution.

**Limitations:**

- The conclusion section includes a thorough treatment of limitations of this work, which helps future work to extend these results. No concerns about negative societal impacts.

---

> ### Author Rebuttal · Authors · 2023-08-09
>
> Thank you very much for reviewing our work. We address your questions and comments below.
>
> > “The main text of the paper does not have an experiment section.” “There is no estimation method or experiment results. Other identifiability papers often contribute an estimation method [...], perform disentanglement experiments in settings that match their theoretical assumptions, or perform ablations on synthetic data where they can control which of their theoretical assumptions are met” “Does your theory suggest any empirical validation that you could add to this paper?”
>
> As our main focus is on identifiability theory, we did not have space for including an experiment section in the main paper for the initial submission. However, we did, in fact, carry out an empirical validation similar to what you suggest. This was initially included in Appendix D (see Fig. 3 and its caption for the main points) and we will use the additional page to move some of this material to the main paper.
>
> In short, we consider an estimation method that involves fitting multiple generative models based on normalizing flows with built-in causal structure for different choices of graphs and intervention targets. Consistent with our theoretical claims, our empirical results show that (i) the correct choice of graph and intervention targets are indeed identified as the ones that yield the best model fit in terms of held-out likelihood; and (ii) the true causal variables are recovered (up to rescaling) as supported by the high MCC values—the standard disentanglement/ICA metric to assess this.
>
> In addition to what is already described in Appendix D, we performed some additional experiments during the rebuttal period, which are summarised in more detail in our general response to all reviewers.
>
> We hope that adding a summary of these (new and old) experiments to the main paper satisfactorily addresses what appeared to be your main concern.
> ____
>
> > Extension of the identifiability result from single interventions to $n>2$
>
> We completely agree that this is desirable. As summarised in l.374-379, we do not see any fundamental reason why this should not be possible, but there are technical obstacles. Thus far, we were not able to find a simple characterisation of the set of genericity conditions required for n>2 variables, but we will continue to investigate this matter.

---

### Author Rebuttal · Authors · 2023-08-09

We thank all reviewers for their feedback and time in reviewing our work. All reviewers recommend acceptance, rating the soundness and contribution of our work as good and its presentation as good or excellent.

We are pleased to read that our paper is “well-motivated and interesting” (`E44d`), ”relevant to causal representation learning”, “more general than prior work” (`FsUF`), “highly enjoyable and insightful”, that it “successfully addresses a challenging and captivating problem”, and “holds significant importance” and “practical relevance” (`U76e`). Reviewers also highlighted that the paper is “well-written” (`FsUF`, `ZLAN`), with `U76e` stating “the organization and writing of this paper are commendable”; and well-situated in the relevant literature (`ZLAN`,`E44d`,`U76e`), “highlight[ing] the unique contributions of [our] work” (`U76e`).

A shared issue raised by reviewers concerns the lack of experiments (in the main paper). To address this, we will use the additional page to add an experiments section to the main paper in the revised version. There we will include a summary of our empirical validation using normalizing flows, which was presented in Appendix D during the initial submission. Moreover, we also conducted some additional experiments during the rebuttal phase, see below and the attached PDF for details and a results figure. We will also add this to the new experiments section.

____


### Summary of New Experiments

**Setting.** Our new experiments extend our initial experiments along several axes:
- more variables: instead of focusing on the setting of Thm. 4.1 with $n=2$, we consider a setting with 3 causal variables and graph $V_2 \leftarrow V_1 \to V_3$

- nonlinear, non-additive noise data-generating process: we use location-scale models instead of linear Gaussian ones for the ground-truth SCMs

- nonparametric causal model: instead of fitting a set of linear parametric conditionals, we use a second normalizing flow (from exogenous variables U to causal variables V) as a nonparametric function approximator of the causal relations

- no enumeration of causal graphs: instead of training a separate generative model for each graph, we only specify the causal ordering (the natural ordering on [n]) and enforce the flow from U to V to have triangular Jacobian, consistent with the causal ordering and acyclic structure; this only leaves an enumeration of the intervention targets.

- violation of assumption (A2’): we consider only one interventional environment per node, thus investigating our conjecture that two paired interventions may not be required for Thm 4.2

**Results.** The results are summarised as a figure in the attached PDF.

(Due to the limited time for hyper-parameter tuning etc, as well as the more challenging training task in this more general setting, the results are more noisy than in the initial experiments. We therefore focus on describing trends.)

We find that the model with correctly specified intervention targets achieves the best fit in terms of likelihood, as well as high MCC. The second-best fit is attained by the $132^{(*)}$ model (yellow violin), for which the targets are misaligned in a way that should, *in principle*, be consistent with the partial permutation ambiguity of the true underlying causal graph. (Note, however, that none of the flow weights are a priori forced to be zero even in the absence of an edge $V_2 \not \to V_3$---this may induce *practical* differences between these two theoretically equivalent models.) This model achieves equally high MCC. All other models (for which the misalignment of intervention targets is incompatible with the partial permutation ambiguity of the true graph) achieve poorer model fits and lower MCC.

**Interpretation.** Overall, these results suggest that enumeration of graphs may not be needed, since 1, …, n is always a valid causal order w.l.o.g., and only choices of intervention targets compatible with the true causal graph achieve the best fits and MCC. Moreover, these results lend empirical support to our conjecture that 1 intervention per node may indeed be sufficient even for n>2 variables.

We will include a more detailed description of the experimental settings and results of these new experiments in the revised manuscript.
____

We address further questions and comments in our responses to the individual reviews.

---

### Decision · Program_Chairs · 2023-09-21

**Decision:**

Accept (poster)

**Comment:**

This paper contributes to the growing literature on identifiability of latent causal models. The authors consider the setting of a nonparametric mixing function, although for > 2 nodes some paired data is required. All reviewers recommend acceptance. The authors should incorporate the changes suggested by the reviewers, for example, more discussion and the promised empirical examples.